

# Probing chaos in the spherical $p$-spin glass model

Lorenzo Correale[1,2][*], Anatoli Polkovnikov[3], Marco Schirò[4] and Alessandro Silva[1]

**1** SISSA — International School for Advanced Studies,
via Bonomea 265, I-34136 Trieste, Italy
**2** INFN — Istituto Nazionale di Fisica Nucleare, Sezione di Trieste, I-34136 Trieste, Italy
**3** Department of Physics, Boston University, 590 Commonwealth Avenue,
Boston, Massachusetts 02215, USA
**4** JEIP, UAR 3573 CNRS, Collège de France, PSL Research University,
11 Place Marcelin Berthelot, 75321 Paris Cedex 05, France

[*] lcorreal@sissa.it

## Abstract

We study the dynamics of a quantum $p$-spin glass model starting from initial states defined in microcanonical shells, in a classical regime. We compute different chaos estimators, such as the Lyapunov exponent and the Kolmogorov-Sinai entropy, and find a marked maximum as a function of the energy of the initial state. By studying the relaxation dynamics and the properties of the energy landscape we show that the maximal chaos emerges in correspondence with the fastest spin relaxation and the maximum complexity, thus suggesting a qualitative picture where chaos emerges as the trajectories are scattered over the exponentially many saddles of the underlying landscape. We also observe hints of ergodicity breaking at low energies, indicated by the correlation function and a maximum of the fidelity susceptibility.

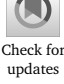

# 1   Introduction

Characterizing the dynamics of a system with multiple equilibrium configurations is a challenging problem encompassing several branches of natural sciences, from biology [1] to economy [2] and theoretical ecology [3]. The nontrivial interplay between stable and unstable configurations makes the dynamical evolution of these systems extremely sensitive to the initial conditions and thus unpredictable: since the pioneering works of the 19th [4,5] and 20th centuries [6,7], this phenomenon has been known as *chaos*.

Among the physical systems displaying multiple equilibria, a prominent role is played by spin-glasses, a class of models originally introduced to describe magnetic alloys [8, 9]: at low temperature, the dynamics is trapped into one of the exponentially many possible metastable states [10–13] for a long-time and explores the phase space through rare, activated jumps between different states [14–16], leading to ergodicity-breaking phenomena [17–20]. The inclusion of quantum fluctuations usually opens up a new route for thermalization in the equilibrium dynamics due to tunnelling between metastable states [21–24], even though counter-intuitive effects of quantum fluctuations inducing glassiness has been found in quantum quench protocols [25] or more recently in presence of more complex energy landscapes [26]. Within such a rich variety of phenomena, the recent theoretical [27–31] and experimental [32] observation of many-body chaos in quantum spin-glasses has drawn particular attention. Specifically, in Refs. [29,30] chaos was detected in the dynamics of a quantum spherical *p*-spin glass model [23], from the exponential growth of an out-of-time-order correlator (OTOC) [33–39]: the corresponding *Lyapunov exponent* $\lambda_L$ exhibits a single broad peak at a temperature scale where the dynamics is still ergodic, but at the onset of slow relaxation.

The behaviour of the Lyapunov exponent $\lambda_L$ extracted from the OTOCs was connected in Ref. [29] to a dynamical crossover between two step and one step time relaxation in the spin-spin correlation functions. This behaviour suggests a deeper connection between this dynamical feature and the structure of the underlying stationary configurations of the model. In particular, if such relation exists other features associated to the peak in other chaos indicators would be observed. Among these chaos indicators, one example is the *Kolmogorov-Sinai* (KS) *entropy* [40–42], that is the sum of all the positive Lyapunov exponents describing growth of the sub-leading terms appearing either in the OTOC for quantum systems [43, 44] or in the distance between nearby trajectories for classical chaotic systems [45]. The KS entropy would indeed provide a deeper insight on the emergence on the strength of chaos and to the entanglement production in a wide range of quantum systems [43, 44, 46–48]. A second, example of indicator focusing on the transition from integrability to chaos in genuine quantum spin systems is the *fidelity susceptibility* [49, 50]. Such quantity, despite being closely connected to the magnetic response obtained in quantum spin-glass experiments [21], has never been investigated in the context of spin-glasses.

The purpose of this work is to develop a qualitative understanding of chaotic behavior in the p-spin model by using phase space methods to compute $\lambda_L$, at fixed energy and in the classical limit. We find that $\lambda_L$ exhibits a broad peak around $E = 0$, at the onset of slow dynamics, and vanishes both at low and high energies, compatibly with previous studies. Working at fixed $E$ we observe that the maximum in $\lambda_L$ occurs when the energy landscape has the maximum possible number of stationary points, as characterized by the complexity (see Ref. [51]). We find that the profile of the Kolmogorov-Sinai entropy is qualitatively identical to that of $\lambda_L$. To further investigate the chaotic features of our model, we also introduce a quantity classically equivalent to the fidelity susceptibility $\chi$ in the ergodic phase finding that, as a function of $E$, it has a single peak, aligning with the onset of non-ergodic behavior in the dynamics. Our results can be easily interpreted in terms of the typical behaviour of the underlying trajectories at different energy scales, which in turn determine the profile of the correlation function: While the low and high-energy limits respectively correspond to trajectories performing small oscillations around a local minimum or a uniform circular motion on the $N$-sphere, for intermediate energies the nontrivial interplay between the saddles of the landscape enhance dynamical chaos. Our method is straightforwardly generalized to other spin-glass models, for example describing spins 1/2 in a transverse field [28], where a similar behaviour for the trajectories is expected both for the low and high-energy limit.

## 2 Classical dynamics of the p-spin glass spherical model

Throughout this work, we will focus on the isolated dynamics of the $p$-spin glass Spherical Model (PSM), whose Hamiltonian

$$\hat{H}_J = \frac{1}{2M} \sum_{i=1}^{N} \hat{\Pi}_i^2 + V_J(\hat{\boldsymbol{\sigma}}), \tag{1}$$

with

$$V_J(\hat{\boldsymbol{\sigma}}) = - \sum_{1 \le i_1 < \cdots < i_p \le N} J_{i_1, \ldots, i_p} \hat{\sigma}_{i_1} \cdots \hat{\sigma}_{i_p}, \tag{2}$$

describes a system of $N$ spins interacting through random, all-to-all couplings $J_{i_1, \ldots, i_p}$, independently sampled from a Gaussian distribution with zero mean and variance $\overline{J^2} = 2p!/N^{p-1}$. Here the spins $\hat{\sigma}_i$ are treated as continuous variables, obeying the spherical constraint $\sum_i \langle \hat{\sigma}_i^2 \rangle = N$ [52], and quantum fluctuations are implemented by the canonical quantization relations $[\hat{\sigma}_i, \hat{\Pi}_j] = i\hbar \delta_{ij}$. The term "spin-glass" is attributed to the quantum PSM model, despite its use of position and momentum operators, due to historical reasons. The introduction of the quantum PSM can be traced back to Ref. [23], where it was revealed to manifest a thermodynamic phase transition from a paramagnetic to a spin-glass state, driven by the temperature $T$ and by a dimensionless parameter $\Gamma = \hbar^2/MJ$, which quantifies the strength of quantum fluctuations. The transition line $\Gamma_c(T)$ that separates these two phases shares qualitative features with the glass transition line experimentally observed in disordered spin 1/2 systems [21,53], coupled to quantum fluctuations through an homogeneous transverse field. The nomenclature "spin-glass" was then associated to the quantum PSM due to these observed similarities. The thermodynamic phase transition in the PSM is either of the first or second order depending on the strength of $\Gamma$ [23,24] and its corresponding transition line can be parametrized also in terms of by a critical temperature $T_c(\hbar)$. At temperature $T_d(\hbar) \gtrsim T_c(\hbar)$, a dynamical, ergodicity-breaking transition is also observed [22], whose properties are sensitive to quenches in $\hbar$ in a non-trivial way [25]. In refs. [29,30], many-body chaos in the quantum PSM was studied using an OTOC in the large-$N$ limit and for a unitary evolution of

the system from a thermal initial state. The OTOC grows exponentially at any temperature $T$, with a *quantum Lyapunov exponent* $\lambda_L(T)$, which displays qualitatively the same profile for a wide range of fixed values of $\hbar$ and in particular in the classical limit $\hbar \to 0$, having a single maximum at $T_m(\hbar) > T_d(\hbar)$ and vanishing in the low and high-temperature limits. In contrast to its fermionic counterpart, the Sachdev-Ye-Kitaev (SYK) model [34,54,55], in the PSM the bound on chaos $\lambda_L \leq 2\pi T/\hbar$, proven for a general quantum many-body system in Ref. [37], is never saturated. A similarity between the Lyapunov exponent of the SYK model and that of the PSM only arises for $\hbar \to 0$, where the bound becomes trivial. In this case, the $\lambda_L(T)$ grows linearly with temperature $T$ at low $T$ for both cases [29,56].

The aim of this work is to gain physical insight on the behaviour of $\lambda_L$ investigating the dynamics of the PSM in the classical limit $\hbar \to 0$ and at fixed energy-density $E$, a framework usually used for classical dynamical systems [42]. This goal can be achieved in the framework of Truncated Wigner Approximation (TWA) [57,58], briefly reviewed in the following (see Refs. [59,60] for further details). We begin by observing that, for a generic system with $N$-dimensional position and momentum-like degrees of freedom, like $\boldsymbol{\sigma}$ and $\boldsymbol{\Pi}$, the Heisenberg dynamics from an initial state $\hat{\rho}$ of any operator $\hat{O} = \mathcal{O}(\hat{\boldsymbol{\sigma}}, \hat{\boldsymbol{\Pi}})$ can be represented as

$$\langle \hat{O}(t) \rangle = \mathrm{Tr}[\hat{\rho}\ \hat{O}] = \int \frac{d\boldsymbol{\sigma}\, d\boldsymbol{\Pi}}{(2\pi\hbar)^N} W_\rho(\boldsymbol{\sigma}, \boldsymbol{\Pi}) O_W(\boldsymbol{\sigma}, \boldsymbol{\Pi}, t), \tag{3}$$

where

$$O_W(\boldsymbol{\sigma}, \boldsymbol{\Pi}, t) = \int d\xi \left\langle \boldsymbol{\sigma} - \frac{\xi}{2} \middle| \mathcal{O}(\hat{\boldsymbol{\sigma}}(t), \hat{\boldsymbol{\Pi}}(t)) \middle| \boldsymbol{\sigma} + \frac{\xi}{2} \right\rangle \exp\left[ i \frac{\boldsymbol{\Pi} \cdot \xi}{\hbar} \right], \tag{4}$$

is called *Weyl symbol* of the operator $\hat{O}$, while the *Wigner function* $W_\rho(\boldsymbol{\sigma}, \boldsymbol{\Pi})$ is the Weyl symbol of the density matrix $\hat{\rho}$. The TWA is then a classical approximation for the dynamics of $O_W(\boldsymbol{\sigma}, \boldsymbol{\Pi}, t)$, summarized as

$$O_W(\boldsymbol{\sigma}, \boldsymbol{\Pi}, t) \simeq \mathcal{O}(\boldsymbol{\sigma}(t), \boldsymbol{\Pi}(t)), \tag{5}$$

where $(\boldsymbol{\sigma}(t), \boldsymbol{\Pi}(t))$ is the classical trajectory evolving from the initial condition $(\boldsymbol{\sigma}, \boldsymbol{\Pi})$ and determined by the following Hamilton equations:

$$\begin{cases} \partial_t \sigma_i = \Pi_i/M\,, \\ \partial_t \Pi_i = -\sum_{j=1}^{N} \left( \delta_{ij} - \frac{\sigma_i \sigma_j}{N} \right) \frac{\partial V_J}{\partial \sigma_j} - \frac{\sum_i \Pi_i^2}{MN} \sigma_i\,. \end{cases} \tag{6}$$

The Eqs. (6) are essentially derived by adding to the Hamiltonian a Lagrange multiplier term $z(t)(\boldsymbol{\sigma}^2 - N)$, where $z(t)$ is determined in a self-consistent way so that the dynamics satisfies the spherical constraint at any time $t$ (see Appendix A for more details). The TWA is believed to be valid at least up to an Ehrenfest time $t_{Ehr} \sim \log \hbar^{-1}$ [33] diverging for small-$\hbar$.

To investigate the dynamics at fixed energy density $E$, the ideal setup would be to fix $\hat{\rho}$ as a micro-canonical state and compute the corresponding Wigner function, which is in general a formidable task. Instead, we fix an initial condition

$$\hat{\rho} = \frac{1}{\mathcal{N}_s} \sum_{l=1}^{\mathcal{N}_s} |\boldsymbol{\alpha}^{(l)}\rangle \langle \boldsymbol{\alpha}^{(l)}|\,, \tag{7}$$

as an ensemble of coherent states $|\boldsymbol{\alpha}^{(l)}\rangle = \otimes_{j=1}^{N} |\alpha_j^{(l)}\rangle$ [61], defined as eigenstates of the operators $\hat{a}_j = \hat{\sigma}_j/(\sqrt{2}l) + il\hat{\Pi}_j/(\sqrt{2}\hbar)$, for $j = 1, \dots, N$ and a free parameter $l$, with eigenvalues

$\boldsymbol{\alpha}^{(l)} = \{\alpha_1^{(l)}, \dots, \alpha_N^{(l)}\}$, respectively. For each state $|\boldsymbol{\alpha}\rangle$, the Wigner function is the Gaussian wave-packet

$$W_{\boldsymbol{\alpha}}(\boldsymbol{\sigma}, \boldsymbol{\Pi}) = 2^N \prod_j \exp\left\{-\frac{(\sigma_j - \sigma_{\boldsymbol{\alpha} j})^2}{l^2} - \frac{l^2(\Pi_j - \Pi_{\boldsymbol{\alpha} j})^2}{\hbar^2}\right\}. \tag{8}$$

In the following, we fix $l = \sqrt{\hbar}$, to have the same uncertainty in the variables $\boldsymbol{\sigma}$ and $\boldsymbol{\Pi}$. The centers $(\boldsymbol{\sigma}_{\boldsymbol{\alpha}}, \boldsymbol{\Pi}_{\boldsymbol{\alpha}})$ of the various wave-packets are chosen in a way that $\langle\boldsymbol{\alpha}|\hat{H}_J|\boldsymbol{\alpha}\rangle / N = E$. For the small values of $\hbar$ we are interested in, it can be easily shown that such choice is equivalent to fix the classical energy density

$$\frac{1}{N}\mathcal{H}_{cl}(\boldsymbol{\sigma}_{\boldsymbol{\alpha}}, \boldsymbol{\Pi}_{\boldsymbol{\alpha}}) = \frac{\boldsymbol{\Pi}_{\boldsymbol{\alpha}}^2}{2MN} + \frac{V_J(\boldsymbol{\sigma}_{\boldsymbol{\alpha}})}{N} \simeq E, \tag{9}$$

so that we can determine the centers of the wave-packets in Eq. (8) using the following "classical annealing" algorithm:

1. First extract a random configuration $(\boldsymbol{\sigma}_0, \boldsymbol{\Pi}_0)$ in phase space, with $\boldsymbol{\sigma}_0$ uniformly sampled on the $N$-sphere and $\boldsymbol{\Pi}_0$ sampled from the normal distribution with zero mean and unit variance.

2. To bring the system in a configuration at the desired energy density $E$, we integrate the dynamics starting from $(\boldsymbol{\sigma}_0, \boldsymbol{\Pi}_0)$ and defined by the equations

$$\begin{cases} \partial_t \sigma_i = \Pi_i/M, \\ \partial_t \Pi_i = -\sum_{j=1}^N \left(\delta_{ij} - \frac{\sigma_i \sigma_j}{N}\right)\left(\frac{\partial V_J}{\partial \sigma_j} + \gamma \Pi_j\right) - \frac{\sum_i \Pi_i^2}{MN} \sigma_i, \end{cases} \tag{10}$$

where a dissipative term of strength $\gamma > 0$ has been added. Notice that $\gamma > 0$ if $\mathcal{H}_{cl}(\boldsymbol{\sigma}_0, \boldsymbol{\Pi}_0) > NE$ and $\gamma < 0$ otherwise.

3. Finally stop the integration as soon as the system reaches a configuration $(\boldsymbol{\sigma}_1, \boldsymbol{\Pi}_1)$ such that $\mathcal{H}_{cl}(\boldsymbol{\sigma}_1, \boldsymbol{\Pi}_1) = NE$. Afterwards we may set $\gamma = 0$ and integrate the Hamilton dynamics (Eq. (6)) from $(\boldsymbol{\sigma}_1, \boldsymbol{\Pi}_1)$ for a time $t_{eq}$, to let the system reach a typical configuration on the corresponding classical microcanonical manifold, which we finally take as the center $(\boldsymbol{\sigma}_{\boldsymbol{\alpha}}, \boldsymbol{\Pi}_{\boldsymbol{\alpha}})$ of the wave-packet in Eq. (4). Throughout the rest of this work, we fix $\gamma = 0.5$ and $t_{eq} = 5$.

In practice, the Wigner function for the initial state in Eq. (7) is obtained taking the average over $\mathcal{N}_s$ different wave-packets, sampled from the classical annealing algorithm for the same fixed configuration of the disorder $\{J_{i_1,\dots,i_p}\}$, as

$$W_\rho(\boldsymbol{\sigma}, \boldsymbol{\Pi}) = \frac{1}{\mathcal{N}_s} \sum_{l=1}^{\mathcal{N}_s} W_{\boldsymbol{\alpha}^{(l)}}(\boldsymbol{\sigma}, \boldsymbol{\Pi}). \tag{11}$$

We repeat the algorithm for each of the $\mathcal{N}_s$ states: the resulting set of points $\{(\boldsymbol{\sigma}_c^{(l)}, \boldsymbol{\Pi}_c^{(l)})\}_{l=1\dots\mathcal{N}_s}$ is then a non-uniform sampling of the classical microcanonical manifold at energy density $E$. As long as $\hbar$ is very small, TWA enables us to sample orbits evolving from a neighborhood of each of each configuration $(\boldsymbol{\sigma}_{\boldsymbol{\alpha}}, \boldsymbol{\Pi}_{\boldsymbol{\alpha}})$, a feature which we will use in Section 3 to investigate classical chaos in the PSM from the average growth of the width of the wave-packets. We compute all the observables we are interested in by averaging over an ensemble of trajectories, evolving according to Eq. (6) from an initial condition $(\boldsymbol{\sigma}, \boldsymbol{\Pi})$ sampled from the distribution in Eq. (11). In the rest of this work, we will denote by $\langle \cdot \rangle$ the average over the sampled trajectories and

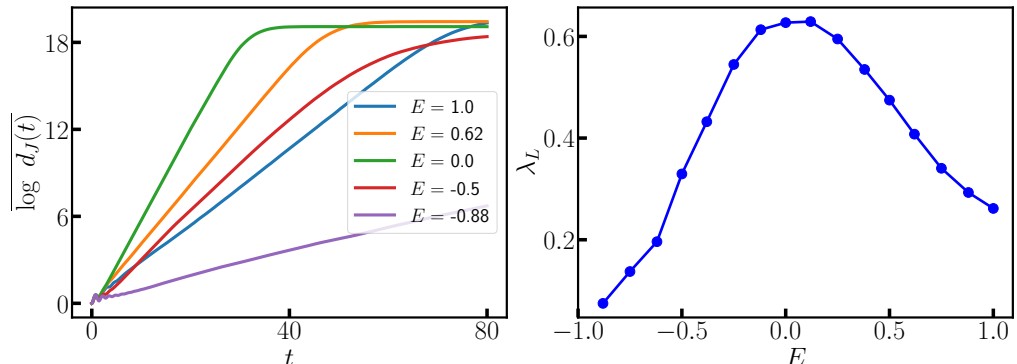

Figure 1: **(Left)** Time-evolution of the log-average over the disorder of the distance $d_J(t)$ in Eq. (12), reported for few paradigmatic energy densities . The average over the initial condition is performed extracting 25 random configurations from each of the $\mathcal{N}_s = 20$ wave-packets, defined in Eq. (8) and obtained through the classical annealing algorithm. The dynamics is integrated up to a time $t_{max} = 80$. The log-average over the disorder is taken over $\mathcal{N}_d = 30$ configurations. **(Right)** Lyapunov exponent $\lambda_L$, defined in Eq. (13), obtained through a linear fit of $\overline{\log d_J(t)}$ over a time window $[t_I, t_F]$, defined in such a way that, for each $E$, $t_I$ is beyond the oscillations at early times displayed by $\overline{\log d_J(t)}$ and $t_F$ is smaller than the saturation time-scale.

by $\overline{(\,\cdot\,)}$ the one over the disorder configurations $\{J_{i_1 \ldots i_p}\}$. We fix $\hbar = 10^{-8}$, $N = 100$ and we focus on the paradigmatic case of $p = 3$.

We conclude this section with two technical remarks. First, it is worth noting that the classical annealing algorithm described above allows us to sample energies that are arbitrarily high, but not arbitrarily low. Specifically, in the PSM, the minima of the potential $V_J(\boldsymbol{\sigma})$ are situated below a characteristic energy scale of $E_{th} = -\sqrt{2(p-1)/p}$ [10] (also discussed in Sec. 4 and detailed in Appendix E). Due to this limitation, our classical annealing approach cannot explore phase space configurations with energies $E < E_{th}$, as the dissipative dynamics of Eq. (10) becomes trapped in the vicinity of the first encountered local minimum. As a result of this constraint on sampling low-energy configurations, we are unable to investigate the relationship between $\lambda_L$ and $E$ in the vicinity of the ground state. Consequently, we are also unable to make a comparison against the linear dependence of $\lambda_L$ on $T$, for small $T$, observed in Ref. [29]. The second remark is that our definition of the Wigner function is not rigorous for the degree of freedom $\boldsymbol{\sigma}$ lying on a compact configuration space (see also Ref. [62]). Specifically, the configurations $\boldsymbol{\sigma}$ sampled from the distribution in Eq. (11) are not confined to the $N$-sphere. Although our approximation fails in capturing precise quantum dynamics at finite $\hbar$, it is expected to be reliable in the limit of $\hbar \to 0$. For our investigative purposes, it serves merely as a tool to examine the classical trajectories evolving from nearby initial configurations sampled at low $\hbar$ (thus testing classical chaos). Furthermore, as $\hbar$ tends to 0, the configurations we extract are expected to asymptotically lie with the $N$-sphere. This is true as long as the center $\boldsymbol{\sigma}_c$ of each sampled wave-packet $W_{\boldsymbol{\alpha}^{(l)}}(\boldsymbol{\sigma}, \boldsymbol{\Pi})$ also resides on the $N$-sphere, a condition consistently met within the classical annealing algorithm.

## 3  Results: Chaos estimators

We present here our results for the chaos estimators of the PSM, starting from the Lyapunov exponent $\lambda_L$. In principle, $\lambda_L$ can be obtained as a function of $E$ by computing the Weyl

symbol, defined in Eq. (4), of the OTOC [28]. However, in the $\hbar \to 0$ limit we are interested in, $\lambda_L$ becomes the exponential rate of divergence of pairs of nearby orbits of the emerging classical dynamics[1] and can be computed from

$$d_J(t) = \frac{1}{2N\hbar} \frac{1}{\mathcal{N}_s} \sum_{l=1}^{\mathcal{N}_s} \sum_{i=1}^{2N} \langle \boldsymbol{\alpha}^{(l)} | \left[ \hat{y}_i(t) - \langle \alpha | \hat{y}_i(t) | \alpha \rangle \right]^2 | \boldsymbol{\alpha}^{(l)} \rangle \,, \tag{12}$$

where $\hat{\mathbf{y}} = (\hat{\sigma}_1, \ldots, \hat{\sigma}_n, \hat{\Pi}_1, \ldots, \hat{\Pi}_N)$ is the set operators corresponding to a classical phase space configuration. The quantity $d_J(t)$ is easier to compute than the OTOC in the TWA framework and, in Appendix B, we show explicitly that both $d_J(t)$ and the OTOC grow exponentially with the same rate in the classical limit. To get rid of the small time scale fluctuations appearing in the dynamics of $\log d_J(t)$, we compute its average $\overline{\log d_J(t)}$ over the disorder, and retrieve a smooth linear growth

$$\overline{\log d_J(t)} \sim \lambda_L t \,, \tag{13}$$

on intermediate time scales, as shown Fig.1 (left). The corresponding Lyapunov exponent $\lambda_L$, shown in Fig. 1 (right) against the energy density $E$, has a clear peak close to $E = 0$, while it asymptotically vanishes at low and high energies. In Refs. [29,30], it was shown that $\lambda_L$ has the same qualitative profile as a function of $T$, displaying for small $\hbar$ a single maximum around $T_m(\hbar) \simeq 1$: this maximum is consistent with the one we find at $E = 0$, since in the paramagnetic phase the classical energy density $E$ and the temperature $T$ are related by the equation $E = T/2 - 1/(2T)$ (as shown in Ref. [63]). Our computed results are also in close numerical agreement with those obtained in Refs. [29,30], where the estimated maximum for the Lyapunov exponent was around $\lambda_L \simeq 0.6$, like in our findings. However, it is important to note that the order of operations in Eq. (13), involving a logarithm and an average over disorder, is reversed compared to previous studies (see Ref. [64] for a more general discussion). Consequently, we do not expect a perfect match between the $\lambda_L$ we compute here and results from Refs. [29,30]. In summary, the exponent $\lambda_L$ we calculate is essentially classical, and the introduction of small fluctuations in the initial conditions is merely a tool we use to sample nearby trajectories starting from the same wave-packet. It is worth mentioning that, while we use quantum fluctuations to sample nearby configurations in phase space, classical chaos can also be probed using different kind of fluctuations (see Ref. [65] for an example).

In the classical limit, the strength of chaos at different energy densities can be also connected to entropy generation by looking at the Kolmogorov-Sinai (KS) entropy [40,41]. This is defined from the observation that, in general, $N$ exponential terms in the form of $\exp(\lambda_i t)$ contribute to the growth of the distance in Eq. (12), with non-negative *Lyapunov exponents* hierarchically ordered as $\lambda_L = \lambda_1 \geq \lambda_2 \geq \cdots \geq \lambda_N \geq 0$ [45,66,67]. The KS entropy (density) is then just the sum $\Lambda_{KS} = \sum_{i=1}^{N} \lambda_i/N$: classically, $\Lambda_{KS}$ quantifies the rate of spreading of coarse-grained volumes in phase space [68], while its quantum counterpart is believed to describe the entanglement growth at early times for a wide range of systems [46–48] and to be a stronger indicator of scrambling dynamics than the single $\lambda_L$ [44]. As shown in Refs. [46,48], the eigenvalues of the symmetric fluctuation matrix

$$\mathbf{G}_{jl}(t) = \frac{1}{2\hbar \mathcal{N}_s} \sum_{l=1}^{\mathcal{N}_s} \langle \left[ \left( \hat{y}_j(t) - \langle \hat{y}_j(t) \rangle \right) \left( \hat{y}_l(t) - \langle \hat{y}_l(t) \rangle \right) \right. \tag{14}$$

$$\left. + \left( \hat{y}_l(t) - \langle \hat{y}_l(t) \rangle \right) \left( \hat{y}_j(t) - \langle \hat{y}_j(t) \rangle \right) \right] \rangle_{\boldsymbol{\alpha}^{(l)}} \,.$$

---

[1]It is important to notice that the classical limit of $\lambda_L$ we are investigating corresponds to *twice* the classical Lyapunov exponent $\lambda_{cl}$ [38], as the classical limit of the OTOC is actually the square of a typical distance between the underlying trajectories (see also Appendix B.1).

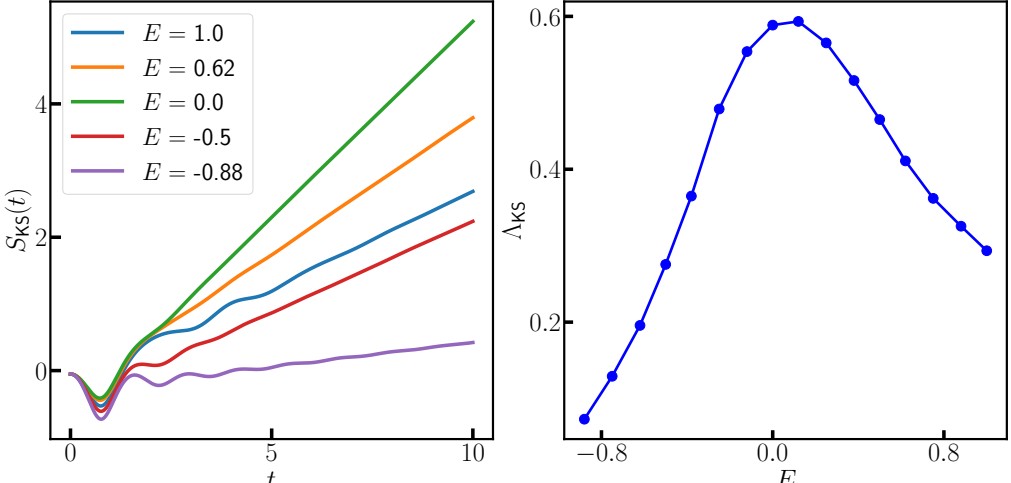

Figure 2: **(Color online) (Left)** Time-evolution of the quantity $S_{\text{KS}}$ defined in Eqs. (14) and (15), reported for few paradigmatic energy densities. The average over the initial condition is performed extracting 30 random configurations from each of the $\mathcal{N}_s = 30$ wave-packets, defined in Eq. (8) and obtained through the classical annealing algorithm. The dynamics is integrated up to a time $t_{max} = 80$. The log-average over the disorder is taken over $\mathcal{N}_d = 96$ configurations. **(Right)** Kolmogorov-Sinai entropy $\Lambda_{\text{KS}}$, defined in Eq. (15), obtained through a linear fit of $S_{\text{KS}}$ over a time window $[t_I, 10]$, where $t_I$ is beyond the scale of oscillations at early time displayed by $S_{\text{KS}}$, for each $E$.

diverge as $\exp(\lambda_i t)$ in the limit $\hbar \to 0$. Then, the KS entropy is straightforwardly extracted from the growth rate of the quantity

$$S_{\text{KS}}(t) = \frac{1}{N} \overline{\log \det[\mathbf{G}_{ij}(t)]_{1 \leq ij \leq N}} \sim \Lambda_{\text{KS}} t \,, \tag{15}$$

on intermediate time-scales. We plot $S_{\text{KS}}(t)$ inf Fig. 2 (left) and, in Fig. 2 (right), its corresponding slope $\Lambda_{\text{KS}}$, showing that the maximal chaos located at $E = 0$ can be detected also by the KS entropy. Notably, a similar result was recently derived also for a classical spin system without disorder [69].

## 4  Chaos, ergodicity and energy landscape

In this section we elaborate further on the results of the previous section and try to provide a qualitative interpretation for the observed maximal chaos around energy $E = 0$ in the PSM. In particular, a natural question is whether this result can be understood from the relaxation dynamics of the PSM at fixed energy density, as probed for example from the spin correlation function, or related to properties of the energy landscape. As we are going to show, the maximal chaos around $E = 0$ in the PSM occurs when the spin relaxation is the fastest and when the complexity of the underlying energy landscape is maximal.

The relaxation dynamics for both classical [17,18] and quantum [22] spin glasses is most often studied in presence of a finite temperature bath. Relaxation is usually defined in terms of the (symmetric) correlation function

$$C(t, t') = \frac{1}{2N} \sum_{i=1}^{N} \overline{\langle \hat{\sigma}_i(t)\hat{\sigma}_i(t') + \hat{\sigma}_i(t')\hat{\sigma}_i(t) \rangle} \,. \tag{16}$$

At high temperatures, the function $C(t_w, t_w + \tau)$ becomes approximately time-translation invariant for moderately high values of $t_w$, and decays to zero for large $\tau$, indicating that the underlying dynamics of the system is ergodic. However, at sufficiently low temperatures, the system may exhibit the so-called 'weak ergodicity breaking scenario' [17, 22, 70] (see Ref. [18] for a review), meaning that

$$\lim_{t_w \to \infty} C(t_w, t_w + \tau) = q_1 + C_{st}(\tau), \tag{17}$$

with a finite dynamical overlap $q_1 > 0$ and where again $C_{st}(\tau)$ vanishes for $\tau \to \infty$, determining a non-ergodic dynamics. In spin glasses, ergodicity breaking is usually accompanied by a breaking of time-translational invariance in $C(t_w, t_w + \tau)$ a phenomenon usually referred to as *aging* [18, 71]: $C(t_w, t_w + \tau)$ has a plateau around $q_1$, whose finite length increases as $t_w$ grows (and diverging for $t_w \to \infty$), before eventually decaying to zero for longer time-scales. Here we are interested instead in the Hamiltonian relaxation dynamics, starting from fixed energy initial conditions. We note that the Hamiltonian dynamics of both classical and quantum PSMs starting from a finite temperature state has been studied recently [25, 63]. To this extent we compute $C(t_w, t_w + \tau)$ in the TWA formalism at finite energy density $E$, making use of Eq. (3) and of the identity

$$\frac{1}{2}\{\hat{\sigma}_i(t)\hat{\sigma}_i(t') + \hat{\sigma}_i(t')\hat{\sigma}_i(t)\}_W = \sigma_i(t)\sigma_i(t'). \tag{18}$$

The results in Fig. 3-(a) show that the correlation function undergoes a temporal crossover from high energies, where it displays wide oscillations, to low energies, where the dynamics slows down. Quite interestingly, at the maximally chaotic point $E = 0$ we observe the fastest relaxation of the correlation function, decaying to zero with few oscillations, again compatibly with Refs. [29, 30]. Upon decreasing further the energy below $E = 0$ the dynamics slows down and we expect a finite plateau around $q_1$ to to appear. Detecting this intermediate plateau within the given simulation time-window is challenging. Consequently, we compute $q_1$ as the value of the correlation function at $t_w = \tau = t_{max}/2$: our results are expected to converge to the ones predicted by Eq. (17) in the limit of $t_{max} \to \infty$. In Figure 3-(b) we show that $q_1$ becomes nonzero below a certain energy threshold, estimated to be around $E_d \simeq -0.38$, and that it increases as $E$ further decreases below the threshold. At the same time, the profiles of the correlation function shown in Fig. 3-(c) lose time-translation invariance again below $E \simeq -0.38$. Notably, these findings are also compatible with the ones obtained from the same simulations performed for a larger $t_{max}$, which are reported in Appendix C and highlight that both a finite $q_1$ and a loss of time-traslational invariance are retrieved at the same energy scale. The results discussed above indicate signs of ergodicity breaking below the energy $E_d \lesssim -0.38$. However, we recognize that our findings, including our estimated value for $E_d$, might undergo quantitative changes with a more extensive analysis using a significantly larger $t_{max}$ than the values considered in this manuscript. Therefore, the ergodicity breaking we observe here should be interpreted as a regime where the dynamics is sufficiently slow for the thermalization time to extend beyond $t_{max}$.

We now argue that both maximal chaoticity and fastest spin relaxation emerge alongside maximal complexity in the topology of the potential $V_J(\boldsymbol{\sigma})$, from Eq. (2), at the $E = 0$ level surface. To understand this connection, we first observe that the profile of the correlation function in Fig. 3-(a) can be associated to the typical behaviour of the underlying trajectories, as sketched in Fig. 3-(d). While the regular oscillations at high energies are due to an underlying uniform circular motion on the $N$-sphere, in the limit of low energies the trajectories oscillate in a well around a local minimum of $V_J(\boldsymbol{\sigma})$, whose amplitude can be roughly estimated as the typical distance between two configurations of the same trajectory, observed at large times

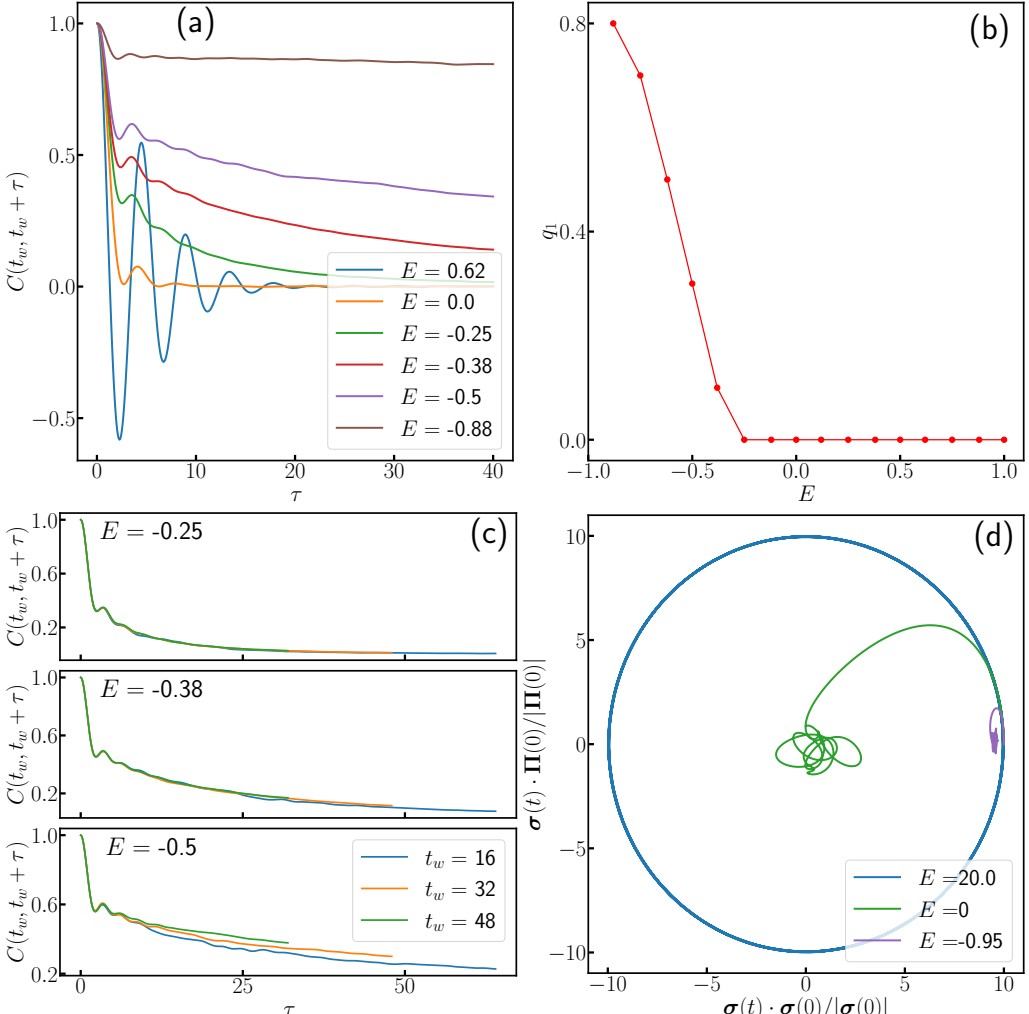

Figure 3: **(a)** Time-dependence of the correlation function, at fixed $t_w = 40$. **(b)** Asymptotic value $q_1$ of the correlation functions $C(t_w, t_w + \tau)$ reported in panel (a), obtained through the time-average of the latter over $\tau \in [39, 40]$. **(c)** Comparison between the profiles of $C(t_w, t_w = \tau)$ obtained fixing different values of $t_w$. Each panel correspond to a different energy density $E$. **(d)** Time-evolution of three typical orbits, whose initial condition are obtained extracting one configuration from the distribution in Eq. (11), at different energy densities $E$ and for the same configuration of the disorder. The orbits evolve in a 200-dimensional phase space and are projected over the two axes defined by the initial spin configuration $\boldsymbol{\sigma}(0)$ and the initial momentum $\boldsymbol{\Pi}(0)$.

separation $\tau$:

$$\sum_{i=1}^{N}[\sigma_i(t_w + \tau) - \sigma_i(t_w)]^2 = 2N - 2\sum_{i=1}^{N}\sigma_i(t_w + \tau)\sigma_i(t_w) \simeq 2N(1 - q_1). \tag{19}$$

Chaos and relaxation emerge in between these two trivial limits, where the trajectories are scattered in neighbors of the stationary configurations of the dynamics in Eq. (6) and explore the whole configuration space. The stationary configurations can be defined as solutions of

the equations (see also Ref. [72])

$$\begin{cases} -\dfrac{\partial V_J}{\partial \sigma_i} + p\dfrac{V_J(\boldsymbol{\sigma})}{N}\sigma_i = 0, \\[2mm] \sum_i \sigma_i^2 = N, \\[2mm] \Pi_i = 0, \end{cases} \tag{20}$$

where in the first equation we used the identity $\sum_j \sigma_j \partial V_J/\partial \sigma_j = pV_J(\boldsymbol{\sigma})$, holding for the potential $V_J(\boldsymbol{\sigma})$ defined in Eq. (2). As discussed in Appendix E the average number of solutions of Eq. (20), lying on the microcanonical manifold at energy density $E = \sum_i \Pi_i^2/2MN + V_J(\boldsymbol{\sigma})/N$, is in a one-to-one correspondence with the stationary points of the potential $V_J(\boldsymbol{\sigma})$ on the $N$-sphere. The average number of such stationary points can then be expressed as (see again Appendix E):

$$\overline{\mathcal{N}(E)} = \int D\sigma \overline{\prod_i \delta\left(-\frac{p}{p!}\sum_{kl} J_{ikl}\sigma_k\sigma_l - pE\sigma_i\right)\left|\det\left(-\frac{p(p-1)}{p!}\sum_k J_{ijk}\sigma_k - pE\delta_{ij}\right)\right|}. \tag{21}$$

For the classical potential $V(\boldsymbol{\sigma})$ in Eq. (2) and in the large-$N$ limit, the number of stationary points scales exponentially as $\mathcal{N}(E) \simeq \exp\{N\Sigma(E)\}$ [10,51], where $\Sigma(E)$ is usually referred to as *complexity*. At the same time, the stability of such stationary points is characterized by the index $k(E)$, where $Nk(E)$ is the average number of unstable directions around every stationary points. In Appendix E, we also derive the analytical expressions for both $\Sigma(E)$ and $k(E)$ as functions of $E$, finding that:

$$\Sigma(E) = \frac{\mathrm{Re}[z(E)]^2 - \mathrm{Im}[z(E)]^2}{p(p-1)} + \frac{1}{2}\log\left\{(\mathrm{Re}[z(E)]^2 - pE)^2 + \mathrm{Im}[z(E)]^2\right\} - E^2 - \frac{1}{2}\log\frac{p}{2} + \frac{1}{2}, \tag{22}$$

where

$$z(E) = \begin{cases} p\left(E + \sqrt{E^2 - 2(p-1)/p}\right)/2, & \text{if } |E| < |E_{th}| \equiv \sqrt{2(p-1)/p}, \\[2mm] p\left(E - \sqrt{E^2 - 2(p-1)/p}\right)/2, & \text{if } |E| > |E_{th}|, \end{cases} \tag{23}$$

and

$$k(E) = \begin{cases} 0, & \text{if } E < E_{th}, \\[2mm] \frac{p}{2\pi(p-1)}E\sqrt{E_{th}^2 - E^2} + \frac{1}{\pi}\arctan\left(\frac{\sqrt{E_{th}^2 - E^2}}{E}\right), & \text{if } |E| < |E_{th}|, \\[2mm] 1, & \text{if } E > |E_{th}|. \end{cases} \tag{24}$$

Intuitively, the value $E_{th} = -\sqrt{2(p-1)/p}$ appearing in the previous formulas represents the *threshold* energy density below which the stationary points are typically local minima of $V(\boldsymbol{\sigma})$, so that $k(E) = 0$ (similarly, $-E_{th}$ is the energy density above which all stationary points of $V(\boldsymbol{\sigma})$ are typically local maxima). Both $\Sigma(E)$ and $k(E)$ are plotted in Fig. 4. The first observation that we make is that the complexity $\Sigma(E)$ has a maximum on the $E = 0$ surface, where stationary points are predominantly saddles of $V_J(\boldsymbol{\sigma})$, surrounded on average by half stable and half unstable directions, as $k(E=0) = 1/2$. Second, we notice that the complexity $\Sigma(E)$ vanishes at two points $E = \pm E_0$, with $E_0 > |E_{th}|$. Beyond $E_0$, we have $\Sigma(E) < 0$, which implies a vanishing number of stationary configurations, so we interrupt the plot at $E_0$. Intuitively, $-E_0$ ($E_0$) can be interpreted as the typical value of the absolute minimum (maximum) of $V_J(\boldsymbol{\sigma})/N$ [72], which is always finite for $\boldsymbol{\sigma}$ lying on the $N$-sphere. We observe that our results for the complexity at fixed energy are compatible with the ones in the literature obtained using different methods [51,73]. The maximum of $\Sigma(E)$ at $E = 0$ unveils a correlation between the

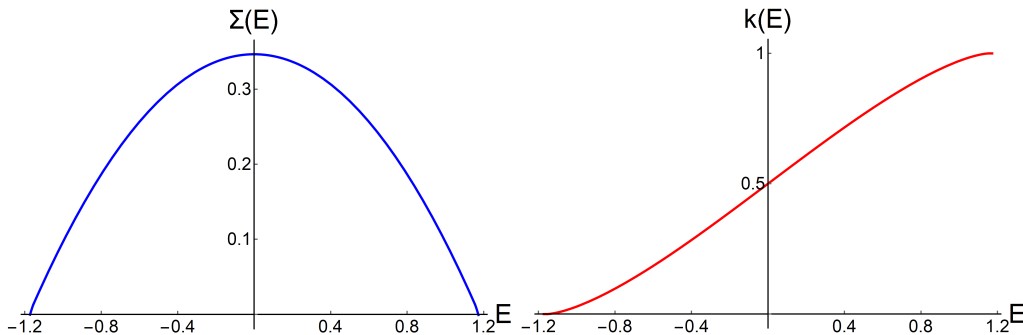

Figure 4: Plots of the complexity function $\Sigma(E)$ (left) and of the average stability index $k(E)$ (right), defined respectively in Eqs.(E.28) and (E.29), for $p = 3$.

number of saddles in the energy landscape and the maximal chaos, detected by $\lambda_L$. This observation suggests that the scattering of trajectories against a maximal number of saddles could offer a potential explanation for the emergence of maximal chaos at $E = 0$.

We conclude our analysis by discussing the connection between chaos and ergodicity in the classical PSM. In the TWA framework we employed, chaos is probed by the Lyapunov exponents while ergodicity is determined by the long-time behavior of the correlation function. However, recent works [50,74] have shown a possibly universal connection between ergodicity in quantum systems, defined there as the emergence of a random matrix behaviour [75], and the *fidelity susceptibility* [49], which characterizes the sensitivity of the eigenstates to an external perturbation. Specifically they shown that an ergodic Hamiltonian exhibits a scaling behavior of $\chi \sim \omega_L^{-1}$ against the mean level spacing $\omega_L$, a behaviour which is consistent with random matrix theory predictions, while for an integrable [76–80] or disorder-localized [80–83] system has a value of order one. Then, a stronger divergence of the fidelity as $\chi \sim \omega_L^{-\alpha}$, with $\alpha > 1$, indicates the approach to a non-ergodic (either integrable or disorder-localized) point. Such scaling is expected to be retrieved in a region around the non-ergodic point which become exponentially small in the system size, when taking the thermodynamic limit. Despite such a complete picture, the fidelity susceptibility has never been tested in classical systems. Here we use it as a tool to identify ergodicity in our $p$-spin model, where ergodicity and its breaking are controlled by the energy density $E$. In particular, here we perturb the Hamiltonian in Eq.(1) with local magnetic fields $B_i$, summarized in the extra term

$$\hat{H}_1 = -\sum_i B_i \hat{\sigma}_i. \tag{25}$$

Then, by perturbation theory, the sensitivity $\chi_n^{(i)}$ of the $n$-th eigenstate to the magnetic field $B_i$ (posing $B_j = 0$ on every other site $j$), is defined by

$$\langle n(0)|n(B_i)\rangle = 1 - \frac{1}{2}\chi_n^{(i)} B_i^2 + O(B_i^3), \tag{26}$$

and we define the fidelity susceptibility as its average $\chi$ over the initial condition in Eq. (7) and the disorder configurations:

$$\chi = \frac{1}{N}\sum_{i=1}^{N}\sum_{n=0}^{\infty}\overline{\langle n|\hat{\rho}|n\rangle \chi_n^{(i)}}. \tag{27}$$

In principle, the fidelity defined in Eq.(27) can be computed classical framework because, as already observed in Ref. [50], $\chi$ can be expressed in terms of the spectral function $\tilde{C}_{av}(\omega)$,

that is the Fourier transform of the time-averaged correlation function

$$C_{av}(\tau) = \lim_{\mathcal{T} \to \infty} \frac{1}{\mathcal{T}} \int_0^{\mathcal{T}} dt_w C(t_w, t_w + \tau). \tag{28}$$

The two are related by the equation (proven in detail in Appendix D):

$$\chi = \int_{|\omega| > \omega_L} \frac{d\omega}{2\pi} \frac{\tilde{C}_{av}(\omega)}{\omega^2}, \tag{29}$$

where $\omega_L$ is again the average quantum level spacing.

In practice, two obstacles prevent us to use directly the Eq.(29). The first one is that we do not have directly access to the spacing $\omega_L$. We then study regularized fidelity (used also in Ref. [74])

$$\chi_\mu = \int \frac{d\omega}{2\pi} \frac{\omega^2}{(\omega^2 + \mu^2)^2} \tilde{C}_{av}(\omega), \tag{30}$$

where we introduced the cutoff $\mu$ to suppress the contribution to the integral from frequencies $|\omega| \lesssim \mu$ and plays a role equivalent to the one played by $\omega_L$ in genuinely quantum systems. We determine the asymptotic profile of $\chi_\mu$ by analyzing its behaviour as $\mu \to 0$. The second, more practical obstacle is that in our simulations we do not have access to infinite-time average, so that the integral in Eq. (28) is performed over a finite time-window $[0, \mathcal{T}]$. While in the ergodic phase this is a good approximation of the long-time average, due to time-translation invariance, in the non-ergodic one $\chi$ always depends on choice of the time-window and converges to the definition in Eq. (27) only in the limit $\mathcal{T} \to \infty$. However, as shown in Appendix D, the qualitative profile we retrieve for $\chi_\mu$ is the same for a wide range of $\mathcal{T}$ between 0 and the maximum integration time $t_{max}$, so that here we can focus on the specific case of $\mathcal{T} = t_{max}/2$.

We plot $\chi_\mu$, as a function of $E$ and for several values of $\mu$, in Fig. 5-(a): its profile has a peak close to the estimated ergodicity breaking energy scale $E \simeq -0.38$ and the maximum point has a little drifting to the left approaching small values of $\mu$. We also observe that, in general, a natural low-frequency cutoff $\Delta\omega \sim 2\pi/t_{max}$ emerges when discretizing the integral in Eq. (30) in our finite-time simulations, so that the asymptotic behaviour of $\chi_\mu$ can be studied only up to $\mu \gtrsim \Delta\omega$. Thus, to refine our analysis, we compute $\chi_\mu$ for a dynamics integrated for a larger $t_{max}$ so that $\mu > 3\Delta\omega$ for all the values of $\mu$ we investigate: the new results, shown in Fig. 5-(b), are qualitatively the same of the one shown in Fig. 5-(a), further validating our analysis. To complete the comparison between our classical analysis and the one performed in Ref. [50], we also analyze the scaling behavior of $\chi_\mu$ against $\mu$ and find that the fidelity susceptibility scales as $\chi_\mu \sim 1/\mu^\alpha$ in the range of values of $\mu$ explored, as shown in Fig. 5-(c). We compute the corresponding exponent $\alpha$ as a function of the energy density $E$ and plot it in Fig. 5-(d): in the ergodic phase, we find that $\alpha$ is slightly greater than 1 (see inset in Fig. 5-(d)), resulting in a scaling approximately consistent with random matrix theory,[2] while $\alpha$ exhibits a maximum of $\alpha \simeq 1.8$ at $E \simeq -0.4$, close to the point where the thermalization time exceeds the simulation time window. Further insight is gained when investigating the profiles of the rescaled fidelities $\mu\chi_\mu$ against $E$, as done in the inset Fig. 5-(d): at high energy densities, the profiles collapse in a region which expands toward lower energies as we decrease $\mu$. This collapse will in general break down at low energies, in particular where the maximum of $\chi_\mu$ is expected to occur. It is also worth mentioning that, from a deeper inspection collapse of the various profiles, we could in principle extract the Thouless time [84], defined as the typical relaxation time-scale $\tau_{th}(E)$ of the correlation function at fixed energy density $E$, using

---

[2]As our analysis is limited by the finite simulation times, we do not exclude that an even better agreement with random matrix predictions may be reached considering a larger $t_{max}$ and consequently a larger set of values for $t_w$.

the following prescription: first we define the cut-off $\mu_E$ as the largest number such that the profiles of $\chi_\mu$ collapse for all energy densities greater than $E$ and for all $\mu < \mu_E$; then the Thouless time can be easily obtained as $\tau_{th}(E) \sim \mu_E^{-1}$ [74]. We expect that $\tau_{th}(E)$ diverges as we the dynamics approaches the ergodicity breaking point from above.

As the cutoff $\mu$ plays the same role of the lowest level spacing for our analysis, these findings are consistent with those in Refs. [50, 74], which show that the fidelity exhibits the strongest divergence with $\mu$ when the corresponding spin relaxation dynamics slows down. We therefore conclude that the fidelity susceptibility could be a good indicator of ergodicity even in classical systems, which warrants further investigation.

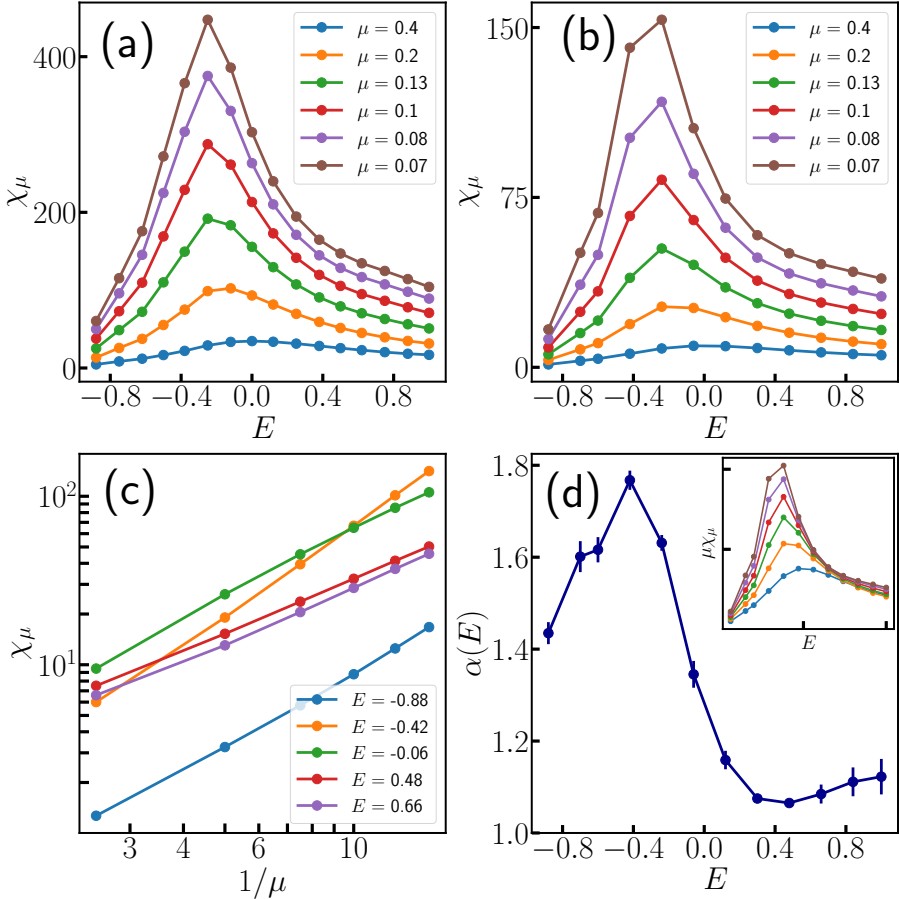

Figure 5: **(Color online)** Fidelity susceptibility $\chi_\mu$ from Eq. (30). **(a)** $\chi_\mu$ is shown as a function of $E$ and fixing several values of the cut-off $\mu$. The data are obtained from a dynamics up to time $t_{max} = 80$, with the same parameters described in Fig. 3. The average time window for the correlation function in Eq. (28) is set to $[0, \mathcal{T}]$, with $\mathcal{T} = t_{max} = 40$. **(b)** Same plot of panel (a), for a dynamics integrated up to time $t_{max} = 320$. Here the average over the initial condition is performed over 5 random configurations extracted from each the $\mathcal{N}_s = 10$ wave-packets constructed by the classical annealing algorithm. We average over $\mathcal{N}_d = 42$ disorder configurations. **(c)** Same data of panel (b). $\chi_\mu$ is shown as a function of the cutoff $\mu$, on a log-log scale, for some fixed values of the energy density $E$. **(d)** Exponent $\alpha(E)$ obtained by a linear fit of $\log \chi_\mu$ against $-\log \mu$, at several fixed values of the energy density $E$. For each $E$, the data used for the fit are the ones from panel (b).

# 5 Conclusions

In this work we have investigated the unitary dynamics of the quantum $p$-spin glass spherical model in the limit of small $\hbar$, where the dynamics is effectively classical, and for the paradigmatic case of $p = 3$. Fixing the initial condition as an ensemble of narrow wave-packets centered on a fixed classical energy shell, at energy density $E$, we have investigated the chaotic dynamics of the model by the exponential divergence of the classical trajectories evolving from the same wave-packets. We have found that the corresponding exponent $\lambda_L$ is maximised when $E$ is close to 0. We have found a similar behavior in a different chaos estimator, the Kolmogorov-Sinai entropy which also shows a pronounced maximum around $E = 0$.

To gain further insights into this result we have investigated the relaxation dynamics of the classical PSM at fixed energy density. We have shown that the spin correlation function displays a crossover in energy, from wide oscillations at high energies to aging low energies, consistently with the known results obtained at finite temperature. Interestingly we have found that around $E = 0$, where the chaos is maximized, the spin relaxation dynamics is the fastest. We give a physical interpretation of all our results in terms of the typical behaviour of the underlying trajectories, which either perform a uniform circular motion at asymptotically high energies or oscillate, at low energies, around a local minimum of the energy landscape. In between these two limits, we suggest that chaos emerges as the trajectories are scattered over the exponentially many saddles of the underlying landscape. Indeed a calculation of the number of stationary configurations shows that the complexity is also maximal at the same energy. Finally, we also gave a classical definition of the fidelity susceptibility. We found that the fidelity has a single maximum, as function of $E$, corresponding to the observed slowing down of the dynamics, a result reminiscent of the ones found in Refs. [50, 74].

The results presented in this study hold true in the $\hbar \to 0$ limit, where the TWA faithfully reproduces the dynamics for the initial condition defined in Eq. (11). Our findings can be potentially extended beyond the realm of small $\hbar$, by utilizing a Wigner state that reproduces the same fluctuations of a realistic quantum micro-canonical or canonical state. In this context, the quantum Lyapunov exponent $\lambda_L$ can be computed even at finite $\hbar$, derived from the exponential growth of a classical analog of the OTOC [28, 33]. A similar rationale applies to the correlation function and subsequently to the fidelity susceptibility, where the latter can be computed using Eq. (30) for both classical and quantum dynamics. Our analysis can also be extended to the transverse-field counterpart of the $p$-spin glass model, where chaos has been recently observed experimentally [32] and where the energy minima exhibit a more complicate structure [85].

# Acknowledgments

We thank L. Cugliandolo and V. Ros for insightful comments. L.C. wants to thank A. Lerose, S. Pappalardi and D. Venturelli for interesting discussions about some of the techniques employed in this work.

**Funding information** M.S. acknowledges support from the ANR grant "NonEQuMat" (ANR-19-CE47-0001). A.P. acknowledges support from the NSF Grant No. DMR-2103658, and AFOSR Grant No. FA9550-21-1-0342.

# A  The truncated Wigner approximation for the p-spin spherical model

In this Appendix we show that, for the quantum spin glass defined in Eq. (1) of the main text in the Truncated Wigner Approximation (TWA) the dynamics is ruled by Eqs.(6) and that such approximation becomes exact in the classical limit. We begin by rewriting the Weyl symbol

$$O_W(\boldsymbol{\sigma}, \boldsymbol{\Pi}, t) = \int d\xi \Big\langle \boldsymbol{\sigma} - \frac{\xi}{2} \Big| \mathcal{O}(\hat{\boldsymbol{\sigma}}(t), \hat{\boldsymbol{\Pi}}(t)) \Big| \boldsymbol{\sigma} + \frac{\xi}{2} \Big\rangle \cdot \exp\left[ i\frac{\boldsymbol{\Pi} \cdot \xi}{\hbar} \right], \qquad (A.1)$$

for a generic operator

$$\mathcal{O}(\hat{\boldsymbol{\sigma}}(t), \hat{\boldsymbol{\Pi}}(t)) = e^{i\hat{H}t} \mathcal{O}(\hat{\boldsymbol{\sigma}}, \hat{\boldsymbol{\Pi}}) e^{-i\hat{H}t}, \qquad (A.2)$$

evolving in the Heisenberg picture. As shown in Ref. [60], the Eq. (A.1) can be represented in a path integral form, suitable to study both the $\hbar \to 0$ and the thermodynamic limit. Without reproducing the details of the calculation, here we just quote the final result:

$$O_W(\boldsymbol{\sigma}, \boldsymbol{\Pi}, t) = \int D\boldsymbol{\sigma} \int D\boldsymbol{\Pi} \int D\xi \int D\boldsymbol{\eta} \, O_W(\boldsymbol{\sigma}(t), \boldsymbol{\Pi}(t))$$

$$\times \exp\Big\{ \frac{i}{\hbar} \int_0^t d\tau \Big[ \boldsymbol{\eta}(\tau) \cdot \frac{\partial \boldsymbol{\sigma}(\tau)}{\partial \tau} - \xi(\tau) \cdot \frac{\partial \boldsymbol{\Pi}(\tau)}{\partial \tau} - \mathcal{H}_W\Big( \boldsymbol{\sigma}(\tau) + \frac{\xi(\tau)}{2}, \boldsymbol{\Pi}(\tau) + \frac{\boldsymbol{\eta}(\tau)}{2} \Big) \quad (A.3)$$

$$+ \mathcal{H}_W\Big( \boldsymbol{\sigma}(\tau) - \frac{\xi(\tau)}{2}, \boldsymbol{\Pi}(\tau) - \frac{\boldsymbol{\eta}(\tau)}{2} \Big) + z(\tau)\xi \cdot \boldsymbol{\sigma} \Big] \Big\},$$

with initial conditions $\boldsymbol{\sigma}(0) = \boldsymbol{\sigma}$, $\xi(0) = \xi$ and $\boldsymbol{\Pi}(0) = \boldsymbol{\Pi}$. The Weyl symbol of the Hamiltonian is defined as $\mathcal{H}_W(\boldsymbol{\sigma}, \boldsymbol{\Pi}) = \boldsymbol{\Pi}^2/2M + V_J(\boldsymbol{\sigma})$, and we inserted a term proportional to the Lagrange multiplier $z(\tau)$ in the action, to enforce the spherical constraint (see also Ref. [22]).

It is straightforward to see that TWA is equivalent to a (classical) expansion of the action in the integrand of Eq. (A.3): indeed we obtain, at leading order, that

$$-\mathcal{H}_W\Big( \boldsymbol{\sigma}(\tau) + \frac{\xi(\tau)}{2}, \boldsymbol{\Pi}(\tau) + \frac{\boldsymbol{\eta}(\tau)}{2} \Big) + \mathcal{H}_W\Big( \boldsymbol{\sigma}(\tau) - \frac{\xi(\tau)}{2}, \boldsymbol{\Pi}(\tau) - \frac{\boldsymbol{\eta}(\tau)}{2} \Big)$$

$$\sim -\xi(\tau) \cdot \frac{\partial \mathcal{H}_W(\boldsymbol{\sigma}(\tau), \boldsymbol{\Pi}(\tau))}{\partial \boldsymbol{\sigma}} + \boldsymbol{\eta}(\tau) \cdot \frac{\partial \mathcal{H}_W(\boldsymbol{\sigma}(\tau), \boldsymbol{\Pi}(\tau))}{\partial \boldsymbol{\Pi}} + O(\boldsymbol{\eta}^2, \xi^2). \qquad (A.4)$$

Integrating out the variables $\boldsymbol{\eta}(\tau)$ and $\xi(\tau)$, we obtain a $\delta$-function constraint on the trajectories $\boldsymbol{\sigma}(\tau)$ and $\boldsymbol{\Pi}(\tau)$:

$$\begin{cases} \dfrac{\partial \boldsymbol{\sigma}}{\partial \tau} = \dfrac{\boldsymbol{\Pi}}{M}, \\[2mm] \dfrac{\partial \boldsymbol{\Pi}}{\partial \tau} = -\dfrac{\partial V_J(\boldsymbol{\sigma}, \boldsymbol{\Pi})}{\partial \boldsymbol{\sigma}} - z(t)\boldsymbol{\sigma}. \end{cases} \qquad (A.5)$$

The Lagrange multiplier is obtained by imposing the constraint $\boldsymbol{\sigma}^2 = N$, that by imposing that the total radial force appearing in the second of Eqs. (A.5) is the correct centripetal one:

$$-\frac{1}{N} \frac{\partial V_J(\boldsymbol{\sigma}, \boldsymbol{\Pi})}{\partial \boldsymbol{\sigma}} \cdot \boldsymbol{\sigma} - z(t) = -\frac{\boldsymbol{\Pi}^2}{MN}. \qquad (A.6)$$

Replacing the expression of $z(t)$ obtained in this way into Eqs. (A.5), we obtain exactly the Eqs. (6) of the main text. Such expansion at leading order is exact in the limit $\hbar \to 0$ where the dynamics is determined by the saddle point of the action appearing in the path-integral.

# B Comparison with exponential growth of the OTOC

In this appendix, we show that, in the classical limit of $\hbar \to 0$, the distance $d_J(t)$ defined in Eq. (12) of the main text diverges with the same Lyapunov exponent $\lambda_L$ of the corresponding out-of-time-order correlator (OTOC).

## B.1 Classical chaos in dynamical systems

For a classical dynamical system, chaos is defined as the exponential divergence in time of the distance between orbits starting at nearby initial conditions. For example, we consider the system of Eqs. (6) of the main text and consider two solutions of such equations, a reference trajectory $\tilde{\mathbf{y}}(\mathbf{y}, t)$ evolving respectively form a generic initial condition $\mathbf{y} = (\sigma, \mathbf{\Pi})$, and a perturbed one $\tilde{\mathbf{y}}(\mathbf{y} + \delta\mathbf{y}, t)$, evolving from $\mathbf{y} + \delta\mathbf{y}$ for $|\delta\mathbf{y}| \ll 1$. Then the reference trajectory is said to be chaotic if the square distance $\Delta(t) = |\tilde{\mathbf{y}}(\mathbf{y} + \delta\mathbf{y}, t) - \tilde{\mathbf{y}}(\mathbf{y}, t)|^2$ grows exponentially in time or, in a more mathematical language, if the corresponding *Lyapunov exponent* [42]

$$\lambda_L = \lim_{t \to \infty} \lim_{\Delta(0) \to 0} \frac{1}{t} \log \frac{\Delta(t)}{\Delta(0)}, \tag{B.1}$$

is positive. In practice, for $\Delta(0) \to 0$ we have that

$$\tilde{\mathbf{y}}(\mathbf{y} + \delta\mathbf{y}, t) - \tilde{\mathbf{y}}(\mathbf{y}, t) \simeq \frac{\partial \tilde{\mathbf{y}}(\mathbf{y}, t)}{\partial \mathbf{y}} \cdot \delta\mathbf{y}, \tag{B.2}$$

so that $\lambda_L$ is also detected by the exponential growth at long times of the matrix elements of the *derivative matrix* $\mathbf{M}(\mathbf{y}, t) = \partial \tilde{\mathbf{y}}(\mathbf{y}, t)/\partial \mathbf{y}$.

We remark that, for classical systems, the Lyapunov exponent $\lambda_{cl}$ is defined from the exponential growth of the distance $\sqrt{\Delta(t)}$, instead of $\Delta(t)$, so that we have $\lambda_L = 2\lambda_{cl}$. However, as explained in the next section, the definition in Eq. (B.1) is the classical limit of the quantum Lyapunov exponent detected by the out-of-time-order correlator (OTOC) and thus more suitable to make a comparison with the chaotic dynamics in a quantum system.

## B.2 The exponential growth of the OTOC

For quantum systems, a generalization of the Lyapunov exponent can be defined from the exponential growth of the average square commutators [33]

$$\mathcal{F}(t) = -\frac{1}{N^2 \hbar^2} \sum_{ij} \langle [\hat{\sigma}_i(t), \hat{\Pi}_j(0)]^2 \rangle, \tag{B.3}$$

retrieved also in the corresponding OTOC [37]

$$\mathcal{C}(t) = \frac{1}{N^2 \hbar^2} \sum_{ij} \langle \hat{\sigma}_i(t) \hat{\Pi}_j(0) \hat{\sigma}_i(t) \hat{\Pi}_j(0) \rangle. \tag{B.4}$$

Such a generalization comes from the observation that, in the limit of small $\hbar$, the commutator $[\hat{\sigma}_i(t), \hat{\Pi}_j(0)]/i\hbar$ can be replaced by the corresponding Poisson parenthesis:

$$\mathcal{F}(t) \simeq \frac{1}{N^2} \sum_{ij} \langle \{\sigma_i(t), \Pi_j(0)\}^2 \rangle = \frac{1}{N^2} \sum_{ij} \langle \left| \frac{\partial \sigma_i(t)}{\partial \sigma_j(0)} \right|^2 \rangle, \tag{B.5}$$

where the quantum average $\langle \cdot \rangle$ is replaced by a suitable average over the classical trajectories. Then, for $\hbar \to 0$, $\mathcal{F}(t)$ is expected to grow with the same Lyapunov exponent $\lambda_L$ characterizing the underlying classical dynamical system.

### B.3 The exponential growth of the distance

In this subsection we show that, for $\hbar \to 0$, the distance $d_J(t)$ defined in Eq. (12) of the main text also grows exponentially in time, with exponent given by $\lambda_L$. We begin by rewriting each of the average over a single wave-packet $|\alpha\rangle$, appearing on the right-hand side of Eq. (12) as

$$\frac{1}{2N\hbar} \sum_i \langle \alpha | \left[ \hat{y}_i(t) - \langle \alpha | \hat{y}_i(t) | \alpha \rangle \right]^2 | \alpha \rangle = \frac{1}{N\hbar} \sum_i \int \frac{d^{2N}y}{(2\pi\hbar)^N} W_\alpha(\mathbf{y}) \left[ \tilde{y}_i(\mathbf{y},t) - \langle \tilde{y}_i(\mathbf{y},t) \rangle_\alpha \right]^2, \quad \text{(B.6)}$$

where we remind that $\tilde{\mathbf{y}}(\mathbf{y},t)$ is a trajectory evolving from $\mathbf{y} = (\sigma, \mathbf{\Pi})$. The average $\langle \cdot \rangle_\alpha$ on the right-hand side of Eq. (B.6) is performed over the coherent wave-packet

$$\frac{W_\alpha(\mathbf{y})}{(2\pi\hbar)^N} = \frac{1}{(\pi\hbar)^N} \exp\left\{ -\frac{(\mathbf{y} - \mathbf{y}_\alpha)^2}{\hbar} \right\}. \quad \text{(B.7)}$$

For $\hbar \to 0$, $W_\alpha(\mathbf{y})$ becomes a $\delta$-function of 0 width. To investigate the behaviour of Eq. (B.6) in this limit, it is useful to perform the change of variable $\mathbf{y} = \mathbf{y}_\alpha + \sqrt{\hbar}\mathbf{x}$. Then we have:

$$\tilde{\mathbf{y}}(\mathbf{y}_\alpha + \sqrt{\hbar}\mathbf{x}, t) \sim \tilde{\mathbf{y}}(\mathbf{y}_\alpha, t) + \sqrt{\hbar}\frac{\partial \tilde{\mathbf{y}}(\mathbf{y}_\alpha, t)}{\partial \mathbf{y}} \cdot \mathbf{x} + O(\hbar),$$

$$\langle \tilde{\mathbf{y}}(\mathbf{y}_\alpha + \sqrt{\hbar}\mathbf{x}, t) \rangle_\alpha \sim \tilde{\mathbf{y}}(\mathbf{y}_\alpha, t) + O(\hbar). \quad \text{(B.8)}$$

Then, plugging Eqs. (B.8) into Eq. (B.6) and performing a little algebra, we obtain

$$\frac{1}{N\hbar} \sum_{i=1}^{2N} \langle \alpha | \left[ \hat{y}_i(t) - \langle \alpha | \hat{y}_i(t) | \alpha \rangle \right]^2 | \alpha \rangle \sim \frac{1}{2N} \sum_{i=1}^{2N} \left| \frac{\partial \tilde{y}_i(\mathbf{y}_\alpha, t)}{\partial y_j} \right|^2 + O(\hbar). \quad \text{(B.9)}$$

As explained in the first subsection of this Appendix, terms in the sum on the right-hand side of Eq. (B.9) growth exponentially in time, for a chaotic system. Equation (B.9) reproduces the dynamics of the derivative matrix only for a finite time-window: the noise produced by the fluctuations of order $\hbar$ eventually lead to a saturation of the typical distance between trajectories evolving from a neighbourhood of $\mathbf{y}_\alpha$. To obtain the Lyapunov exponent within such finite time-scale, we also average over the possible configurations of $\mathbf{y}_\alpha$ on the same energy shell, as explained in the main text and obtain

$$d_J(t) = \frac{1}{2N\hbar} \frac{1}{\mathcal{N}_s} \sum_{\alpha=1}^{\mathcal{N}_s} \sum_{i=1}^{2N} \langle \alpha | \left[ \hat{y}_i(t) - \langle \alpha | \hat{y}_i(t) | \alpha \rangle \right]^2 | \alpha \rangle, \quad \text{(B.10)}$$

which for small $\hbar$ is expected to grow exponentially with exponent $\lambda_L$, as confirmed also by Fig. 1 of the main text.

## C Correlation function on a longer time-scale

In this Appendix, we discuss some results obtained from the dynamics of the $p$-spin model on larger time-scales. In particular, we integrate the dynamics according to the protocol described in Section 2 of the main text, and compute the corresponding correlation function $C(t, t')$ in Eq. (16). Here the averages are performed on fewer realizations, with respect to the results presented in the main text. The results in Fig. 6 show that the correlation function seems to break time-translation invariance on energy scales between $-0.25$ and $-0.42$, even though the profiles of $C(t_w + \tau, t_w)$, for various $t_w$, are less smooth due to the noise induced by the fewer realization we took. The results shown here are anyway compatible with the ones presented in the main text. The results shown in Fig. 6 are anyway compatible with the ones presented in the main text and are also qualitatively similar with the plots describing the correlation function of a classical spin-glass in the non-ergodic phase and in the large-$N$ limit, see for example Fig. (10) of Ref. [18].

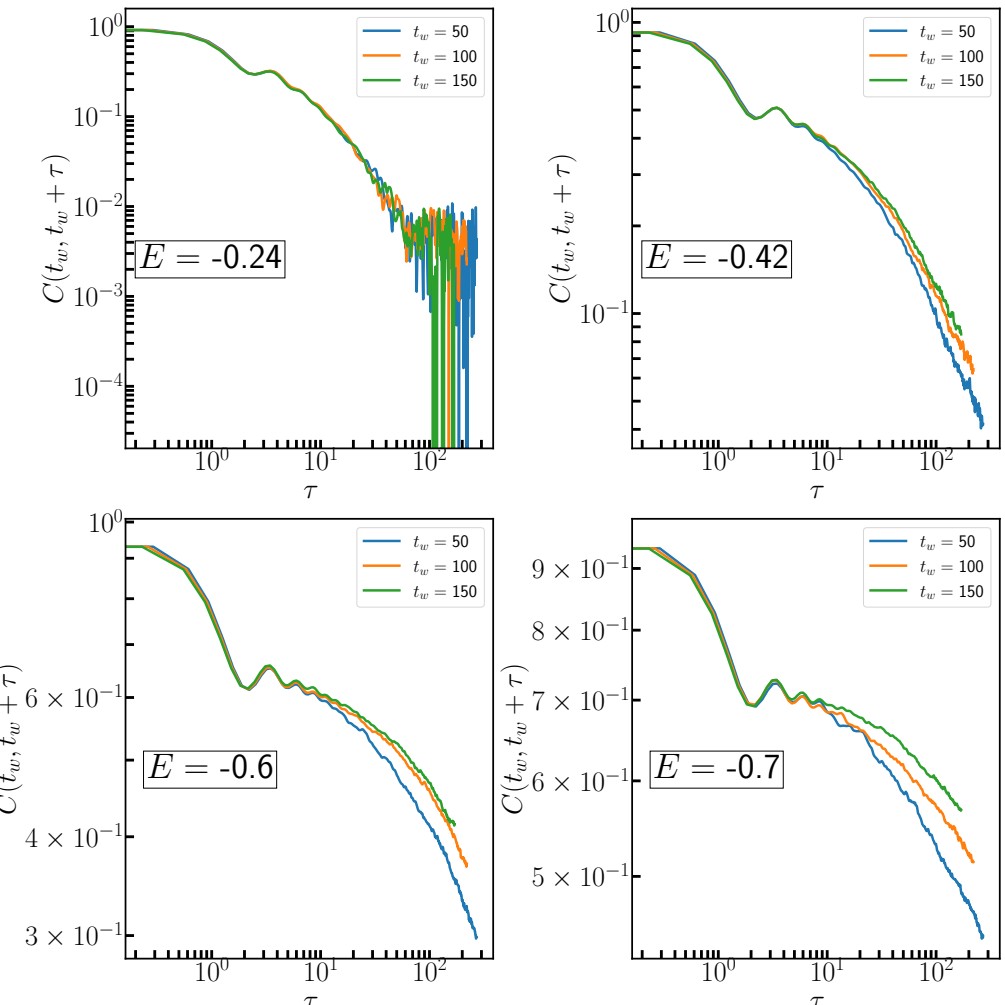

Figure 6: Several plots of the correlation function $C(t_w + \tau, t_w)$, for different fixed values of the energy density $E$. For all the panels, the data are obtained from a dynamics up to time $t_{max} = 320$, with the same simulation parameters described in Fig. 5-(b) of the main text. The data are presented on a log-log scale.

# D  Details on the fidelity susceptibility

In this appendix, we prove the expression in Eq. (27) of the main text for the average fidelity susceptibility $\chi$. We also discuss how the qualitative profile of the fidelity $\chi_\mu$, defined in Eq. (30) and shown in Figg. 5-(a) and (b) of the main text, is independent both of the time window $[0, \mathcal{T}]$, over which we average the average correlation function in Eq. (28), and of the value of the cut-off $\mu$, provided that the latter is sufficiently small.

We begin by recalling our definition for the fidelity susceptibility: We perturb the Hamiltonian $\hat{H}_J = \frac{1}{2M} \sum_{i=1}^{N} \hat{\Pi}_i^2 + V_J(\hat{\sigma})$ of the PSM with local magnetic fields, as

$$\hat{H}(\boldsymbol{B}) = \hat{H}_J - \sum_{i=1}^{N} B_i \hat{\sigma}_i, \tag{D.1}$$

and define the local susceptibilities of the as

$$\chi_n^{(i)} = \left[ -\frac{\partial^2}{\partial B_i^2} \langle n | n(\boldsymbol{B}) \rangle \right]_{\boldsymbol{B}=0}, \tag{D.2}$$

where $|n(\boldsymbol{B})\rangle$ is the $n$-th eigenstate of the perturbed Hamiltonian $\hat{H}(\boldsymbol{B})$ and $|n\rangle = |n(0)\rangle$. By standard calculations of non-degenerate perturbation theory [86], we have

$$\chi_n^{(i)} = \sum_{m \neq n} \frac{|\langle n|\hat{\sigma}_i|m\rangle|^2}{(E_n - E_m)^2}, \tag{D.3}$$

where $E_n$ is the $n$-th unperturbed energy level of the Hamiltonian $\hat{H}_J$. For the initial state $\hat{\rho}$ defined in Eq. (7) of the main text, we define the fidelity susceptibility $\chi$ as the weighted average

$$\chi = \frac{1}{N} \sum_{i=1}^N \sum_n \overline{\langle n|\hat{\rho}|n\rangle \, \chi_n^{(i)}} = \frac{1}{N} \sum_{i=1}^N \sum_{n, m \neq n} \overline{|\langle n|\psi\rangle|^2 \frac{|\langle n|\hat{\sigma}_i|m\rangle|^2}{(E_n - E_m)^2}}, \tag{D.4}$$

performed over the sites, the eigenstates and the disorder configurations. Then, we observe that $\chi$ is connected to the Fourier transform of the time-averaged correlation function

$$C_{av}(\tau) = \lim_{\mathcal{T} \to \infty} \frac{1}{\mathcal{T}} \int_0^{\mathcal{T}} dt_w C(t_w, t_w + \tau). \tag{D.5}$$

In particular, $C(t_w, t_w + \tau)$ and $C_{av}(\tau)$ can be represented as follows:

$$C(t_w, t_w + \tau) = \frac{1}{N} \sum_{i=1}^N \sum_{lmn} \overline{e^{i(E_l - E_n)t_w} e^{i(E_l - E_m)\tau} \langle l|\hat{\sigma}_i|m\rangle \langle m|\hat{\sigma}_i|n\rangle \langle \psi|l\rangle \langle n|\psi\rangle},$$
$$C_{av}(\tau) = \frac{1}{N} \sum_{i=1}^N \sum_n \sum_m \overline{|\langle n|\psi\rangle|^2 e^{i(E_n - E_m)\tau} |\langle n|\hat{\sigma}_i|m\rangle|^2}. \tag{D.6}$$

The second line leads immediately to the Lehmann representation of the average correlation function, which reads as[3]

$$\tilde{C}_{av}(\omega) = \int_{-\infty}^{\infty} d\tau e^{-i\omega\tau} C_{av}(\tau) = \frac{2\pi}{N} \sum_{i=1}^N \sum_{nm} \overline{\langle n|\hat{\rho}|n\rangle |\langle n|\hat{\sigma}_i|m\rangle|^2 \delta(\omega - E_n + E_m)}. \tag{D.7}$$

The latter is immediately related to the typical susceptibility $\chi$ in Eq.(D.4) via the expression

$$\chi = \int_{|\omega| > \omega_L} \frac{d\omega}{2\pi} \frac{\tilde{C}_{av}(\omega)}{\omega^2}, \tag{D.8}$$

where $\omega_L$ is the average spacing of the unperturbed energy levels [50].

As stated in the main text, within the TWA framework we can not compute the fidelity directly from Eq. (D.8) for two reasons: first, our simulations are performed up to a finite maximum time $t_{max}$, so that we can perform the average in Eq. (28) only on a finite time window $[0, \mathcal{T}]$, with $\mathcal{T} < t_{max}$; second, in the limit $\hbar \to 0$ we do not have access to the spacing $\omega_L$ and have a frequency cut-off set by $\Delta\omega = 2\pi/t_{max}$. The second issue was already solved by using the regularized fidelity $\chi_\mu$, from Eq. (30) of the main text, in place of $\chi$. In Fig. 7 we also show that the profile of $\chi_\mu$ is qualitatively the same for a wide range of $\mathcal{T}$ between 0 and $t_{max}$, so that the first issue is actually irrelevant for our results.

---

[3]In our framework, $C(t_w\tau, t_w)$ is actually defined only for $\tau > t_w$. To compute the Fourier transform, we first symmetrized $C_{av}(\tau)$ with respect to $\tau$.

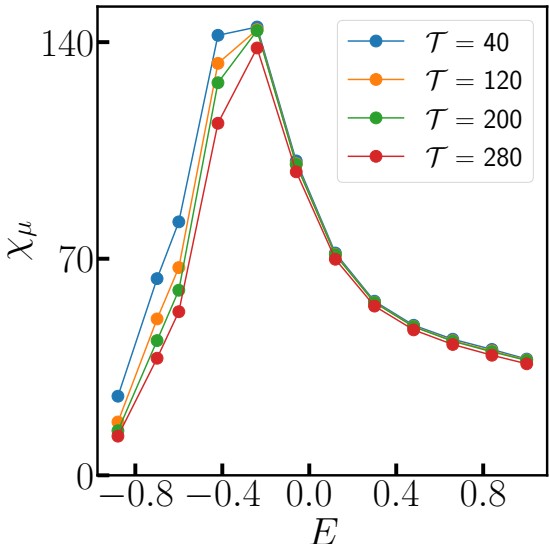

Figure 7: Fidelity susceptibility $\chi_\mu$ from Eq. (30) of the main text, shown as a function of $E$ for a fixed cutoff $\mu = 0.07$. The average time window for the correlation function in Eq. (28) is set to $[0, \mathcal{T}]$, for several values of $\mathcal{T}$. The data are obtained from a dynamics up to time $t_{max} = 320$, with the same parameters described in Fig. 5-(b) of the main text.

## E  Calculation of the complexity

In this appendix, we compute in detail the average number of stationary configurations of the dynamics induced by Eq. (6) of the main text. We start by rewriting the Eqs. (20), defining such stationary points:

$$\begin{cases} -\dfrac{\partial V_J}{\partial \sigma_i} + p\dfrac{V_J(\boldsymbol{\sigma})}{N}\sigma_i = 0, \\ \displaystyle\sum_i \sigma_i^2 = N, \\ \Pi_i = 0. \end{cases} \tag{E.1}$$

More precisely, we look for a solution of Eq. (E.1) lying on a microcanonical manifold defined by the equation

$$E = \frac{\sum_i \Pi_i^2}{2MN} + \frac{V_J(\boldsymbol{\sigma})}{N}. \tag{E.2}$$

Thus, as the kinetic energy vanishes (as $\Pi_i = 0$ for every $i$), we can rewrite the equations for the stationary configurations in the following equivalent form:

$$\begin{cases} -\dfrac{\partial V_J}{\partial \sigma_i} + p\dfrac{V_J(\boldsymbol{\sigma})}{N}\sigma_i = 0, \\ \dfrac{V_J(\boldsymbol{\sigma})}{N} = E, \\ \displaystyle\sum_i \sigma_i^2 = N, \\ \Pi_i = 0. \end{cases} \tag{E.3}$$

The system of Eqs. (E.3) can be further simplified by the following two observations. The first one is that, as we are interested in counting the number of the solutions of Eqs. (E.3), we

can just focus on the number of solutions of the equations involving $\boldsymbol{\sigma}$, as the trivial equations $\Pi_i = 0$ do not bring any degeneracy. The second one is that the equations involving $\boldsymbol{\sigma}$, written in the first three lines of Eqs. (E.3), are equivalent to the following reduced system of equations

$$\begin{cases} -\dfrac{\partial V_J}{\partial \sigma_i} + p\dfrac{E}{N}\sigma_i = 0\,, \\ \sum_i \sigma_i^2 = N\,. \end{cases} \tag{E.4}$$

The equivalence can be seen by multiplying the first line of the system of Eqs. (E.4) by $\sigma_i$ and summing over $i$: by making use of the spherical constraint, one recovers the equation $E = V_J(\boldsymbol{\sigma})/N$; then, by substituting back $E = V_J(\boldsymbol{\sigma})/N$ in the first line of Eqs. (E.4), we obtain the first line of Eqs. (E.3).

To summarize, the number of stationary points of Eqs. (20) lying on a manifold at energy density $E$ coincides with the number of solution of the equations

$$-\frac{\partial V_J}{\partial \sigma_i} + p\frac{E}{N}\sigma_i = 0\,, \tag{E.5}$$

for $i = 1, \ldots, N$, lying on the $N$-sphere. In the spirit of Ref. [72], in what follows we will often write the indices for the $p = 3$ case, such that $J_{i1\ldots ip}$ becomes $J_{ijk}$. However, to give formulas that are valid even in the general case, we will write all the factors containing a term $p$ for the generic $p$. Then, the average number of solutions of Eq. (E.5) on the $N$-sphere then reads:

$$\overline{\mathcal{N}(E)} = \int D\sigma \overline{\prod_i \delta\left(-\frac{p}{p!}\sum_{kl} J_{ikl}\sigma_k\sigma_l - pE\sigma_i\right)\left|\det\left(-\frac{p(p-1)}{p!}\sum_k J_{ijk}\sigma_k - pE\delta_{ij}\right)\right|}\,, \tag{E.6}$$

the overbar denoting the average over the disorder and the spherical constraint will be from now on hidden in the integration measure $D\sigma = \delta(\sum_i \sigma_i^2 - N)\prod_{i=1}^N d\sigma_i$. The absolute value of the determinant appearing on the right-hand side is just a Jacobian factor appearing because the Eqs. (E.5) are written in an implicit form.

To compute $\overline{\mathcal{N}(E)}$ in the large-$N$ limit, we need two approximations. The first one consists in assuming that there is no correlation between the last two terms in the right-hand side of Eq. (E.6) [11,72], that is

$$\overline{\delta\left(-\frac{p}{p!}\sum_{kl} J_{ikl}\sigma_k\sigma_l - pE\sigma_i\right)\left|\det\left(-\frac{p(p-1)}{p!}\sum_k J_{ijk}\sigma_k - pE\delta_{ij}\right)\right|}$$
$$\simeq \overline{\delta\left(-\frac{p}{p!}\sum_{kl} J_{ikl}\sigma_k\sigma_l - pE\sigma_i\right)}\;\overline{\left|\det\left(-\frac{p(p-1)}{p!}\sum_k J_{ijk}\sigma_k - pE\delta_{ij}\right)\right|}\,, \tag{E.7}$$

so that each the average of the $\delta$-function and the one of the determinant can be computed independently from each other. We compute the average $\delta$-function by using the exponential representation

$$\overline{\delta\left(-\frac{p}{p!}\sum_{kl} J_{ikl}\sigma_k\sigma_l - pE\sigma_i\right)} = \int \prod_i \frac{d\mu_i}{2\pi} \overline{e^{-ip/p!\sum_{ikl} J_{ikl}\mu_i\sigma_k\sigma_l}} e^{ipE\sum_j \mu_j\sigma_j}\,, \tag{E.8}$$

and averaging out the disorder after a proper symmetrization of the exponent. Taking into account the spherical constraint $\sum_i \sigma_i^2 = N$ and posing $J = 1$ for simplicity, we get

$$\overline{\delta\left(-\frac{p}{p!}\sum_{kl} J_{ikl}\sigma_k\sigma_l - pE\sigma_i\right)} = \int \prod_i \frac{d\mu_i}{2\pi} \exp\left\{-\frac{p}{4}\sum_j \mu_j^2 - \frac{p(p-1)}{4N}(\sum_j \mu_j\sigma_j)^2 + ipE\sum_j \mu_j\sigma_j\right\}\,. \tag{E.9}$$

To get rid of the term $(\sum_j \mu_j \sigma_j)^2$, we perform and extra Hubbard-Stratonovich transformation and perform straightforwardly all the remaining Gaussian integrals, posing again $\sum_i \sigma_i^2 = N$ in all our expressions. The final result is:

$$\overline{\delta\left(-\frac{p}{p!}\sum_{kl} J_{ikl}\sigma_k\sigma_l - pE\sigma_i\right)} \simeq \frac{1}{(2\pi)^{N/2}}\exp\left\{-N\left(E^2 + \frac{1}{2}\log\frac{p}{2}\right)\right\}, \tag{E.10}$$

up to a multiplicative constant which becomes irrelevant in the thermodynamic limit. The integration of the Jacobian factor is a bit more tricky. To perform it, we make our second approximation by assuming that the sign of the determinant, for any fixed configuration of the disorder, is given by the average number of negative eigenvalues of the corresponding Hessian matrix at energy density $E$, that we write as $Nk(E)$. In formulas, this is equivalent to:

$$\overline{\left|\det\left(-\frac{p(p-1)}{p!}\sum_k J_{ijk}\sigma_k - pE\delta_{ij}\right)\right|} \simeq \overline{\det\left(-\frac{p(p-1)}{p!}\sum_k J_{ijk}\sigma_k - pE\delta_{ij}\right)}\cdot(-1)^{-Nk(E)}, \tag{E.11}$$

$k(E)$ being the average *fraction* of negative eigenvalues. Once we got rid of the modulus, we rewrite the average of the determinant using a fermionic representation [87]:

$$\overline{\det\left(-\frac{p(p-1)}{p!}\sum_k J_{ijk}\sigma_k - pE\delta_{ij}\right)} = \int\prod_j d\psi_j d\overline{\psi}_j \overline{e^{-p(p-1)/p!\sum_{ikl}J_{ikl}\sigma_i\overline{\psi}_k\psi_l}}e^{-pE\sum_i\overline{\psi}_i\psi_i}. \tag{E.12}$$

Integrating out the disorder and using again an Hubbard-Stratonovich transformation to get rid of quartic fermionic terms (see Ref. [72] for more details about this step), we arrive at

$$\overline{\det\left(-\frac{p(p-1)}{p!}\sum_k J_{ijk}\sigma_k - pE\delta_{ij}\right)} \propto \int_{-i\infty}^{i\infty} dz\, e^{NG(z)}, \tag{E.13}$$

where $G(z) = \frac{z^2}{p(p-1)} + \log(z - pE)$. Plugging everything together, we have that (up to an irrelevant prefactor)

$$\overline{\mathcal{N}(E)} = (-1)^{Nk(E)}\exp\left\{-N\left(E^2 + \frac{1}{2}\log\frac{p}{2}\right)\right\}\int_{-i\infty}^{i\infty} dz\, e^{NG(z)}. \tag{E.14}$$

Thus, we are left with the computation of the integral

$$I_\Gamma = \int_\Gamma dz\, e^{NG(z)}, \tag{E.15}$$

along the imaginary axis $\Gamma$ in the complex plane. In the thermodynamic limit $N \to \infty$, this goal can be achieved by using the saddle-point method [88], which we briefly review in the following. First, we observe that, for generic $z = x + iy$ (for $x, y$ real numbers), the function $G(z) = u(x,y) + iv(x,y)$ can be decomposed in its real and imaginary parts as

$$\begin{cases} u(x,y) = \frac{1}{2}\log[(x-pE)^2 + y^2] + \frac{x^2 - y^2}{p(p-1)}, \\ v(x,y) = \frac{2}{p(p-1)}xy + \arctan\frac{y}{x-pE} + \pi\Theta(pE-x)\text{sign}(y). \end{cases} \tag{E.16}$$

In summary, the saddle point method states that if we find a deformation $\gamma$ of $\Gamma$ in the context plane such that:

1. $v(x, y)$ is constant over $\gamma$,

2. $u(x, y)$ has a global maximum along $\gamma$ at some point $z = z_0$,

3. $G(z)$ is analytic in the closed domain encompassed by the curves $\Gamma$ and $\gamma$,

we have that

$$I_\Gamma = I_\gamma \equiv \int_\gamma dz\, e^{NG(z)} \simeq \exp\left(NG(z_0) + o(N)\right), \tag{E.17}$$

where the last asymptotic relation holds in the $N \to \infty$ limit and is known as Laplace method [88]. It is easy to see that the first condition is equivalent to state that $\gamma$ is parallel to $\nabla u(x, y)$, as the relation $\nabla u \cdot \nabla v = 0$ holds for the holomorphic function $G(z)$. Then if $\nabla u(x, y)$ vanishes along $\gamma$, it vanishes in the whole $\mathbb{R}^2$ plane, so that any maximum of $u(x, y)$ along $\gamma$ is a stationary point of $u(x, y)$ in $\mathbb{R}^2$.

However, the choice of a suitable $\gamma$ depends on the value of the energy density $E$, because $G(z)$ has a branch-cut on the half-line $\{z = x | x < pE\}$ (see the expression of $v(x, y)$ in eq (E.16)) and the position in the complex plane of the stationary points of $u(x, y)$, given by

$$z_\pm(E) = \frac{p}{2}\left(E \pm \sqrt{E^2 - \frac{2(p-1)}{p}}\right), \tag{E.18}$$

also depends on $E$. In particular, we can identify four relevant energy windows, which we treat separately:

$$\begin{aligned} E < E_{th} \equiv -\sqrt{2(p-1)/p}, \qquad & E_{th} < E < 0, \\ 0 < E < |E_{th}|, \qquad & E > |E_{th}|. \end{aligned} \tag{E.19}$$

For $E < E_{th}$, we can take the curve $\gamma$ as the level curve $v(x, y) = v(z^+(E)) = 0$[4] (shown in Fig. 8-(a)). It is straightforward to show that $\gamma$ is the only deformation of $\Gamma$ along which $u(x, y)$ displays a maximum, located at $z = z_+(E)$. Thus for $N \to \infty$ we have $I_\Gamma \simeq \exp\{NG(z_+(E))\}$ and $\overline{\mathcal{N}(E)} \simeq \exp\{N\Sigma(E)\}$, where

$$\Sigma(E) = \frac{z_+(E)^2}{p(p-1)} + \log(z_+(E) - pE) - E^2 - \frac{1}{2}\log\frac{p}{2} + \frac{1}{2}, \tag{E.20}$$

where we posed the phase $k(E) = 0$ in Eq. (E.14), as $\overline{\mathcal{N}(E)}$ has to be a positive real number. The physical meaning of having $k(E) = 0$ is that, in this energy range, the integral in Eq. (E.6) is dominated by local minima of the potential $V_J(\boldsymbol{\sigma})$, where the Hessian is positive definite.

For $E_{th} < E < 0$, the only suitable deformation of $\Gamma$ is $\gamma = \gamma_+ \cup \gamma_-$, where $\gamma_+$ and $\gamma_-$ are respectively the two level curves $v(z) = v(z_+(E))$ and $v(z) = v(z_-(E))$ of $v(z)$. The two curves intersect respectively the points $z_+(E)$ and $z_-(E)$ in the complex plane (see Fig. 8-(b)), which in turn are maxima of $u(x, y)$ along each of the two curves. Then, as $u(z_+) = u(z_-)$ and $v(z_+) = -v(z_-)$, the $N \to \infty$ asymptotic value of $I_\Gamma$ is the sum of two contributions:

$$I_\Gamma \simeq \frac{e^{iNv(z_+)} + e^{-iNv(z_+)}}{2} e^{Nu(z_+)}. \tag{E.21}$$

To give a physical interpretation to our result, let us first note that the function $Nk(E)$, defined in Eq. (E.11), is a non-negative integer. This is because $Nk(E)$ is the average number of

---

[4]To simplify notation, we will abuse notation by writing $v(z)$ to represent $v(\mathrm{Re}(z), \mathrm{Im}(z))$, and similarly for $u(x, y)$. This allows us to write equations more compactly and avoid cluttering them with repetitive expressions.

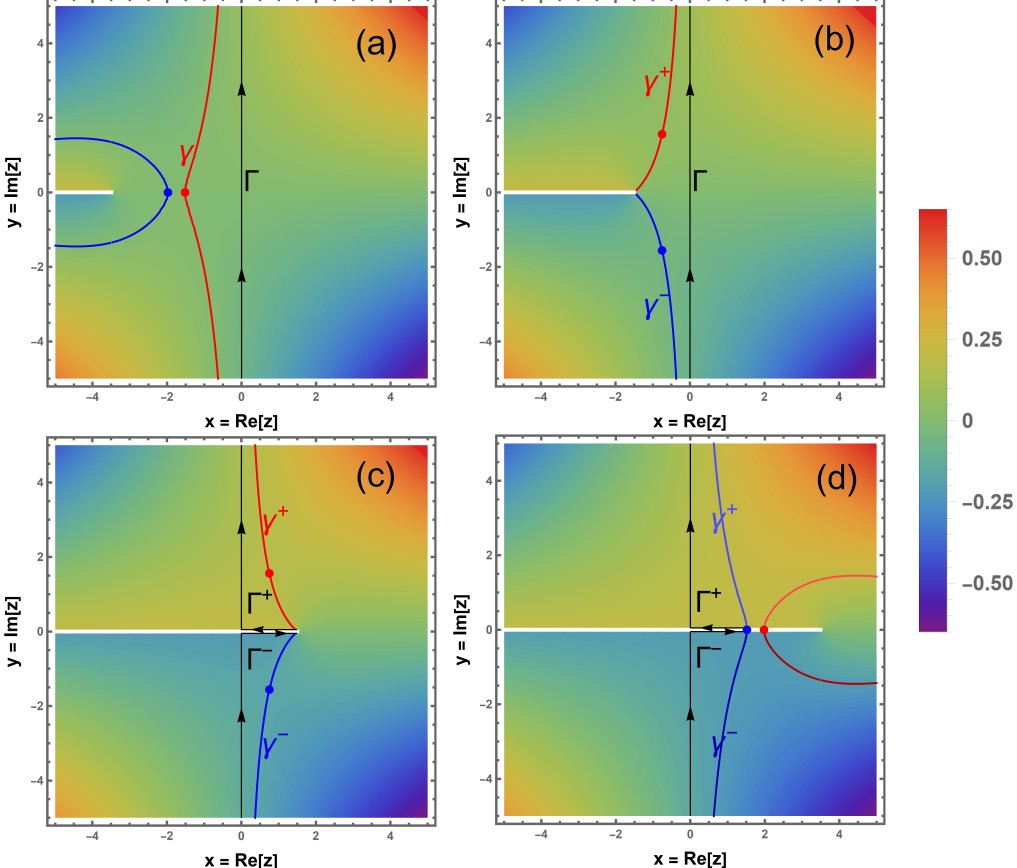

Figure 8: **(Color online)** Color map in the complex plane representing the values assumed by the function $v(x, y)$ defined in the appendix. Each panel corresponds to a different, fixed value of the energy density $E$, each representative of one of the four, qualitatively different cases studied in the appendix. Here we plot the results for $p = 3$. The white thick line correponds to the branch-cut of $v(z) = v(x, y)$ in the complex plane, where $z = x + iy$, while the black lines represent the integration contour $\Gamma$ or its deformation $\Gamma^+ \cup \Gamma^-$. The blue and the red lines correspond to the level curve of $v(z)$, passing respectively through $z^-(E)$ and $z^+(E)$. **(a)** $E < E_{th}$. **(b)** $E_{th} < E < 0$. **(c)** $0 < E < |E_{th}|$. **(d)** $E > |E_{th}|$.

negative eigenvalues of the Hessian matrix associated with $V_J(\sigma)$. Thus, even though the values of $k(E)$ become dense in the interval $[0, 1]$ as $N$ approaches infinity, at any finite $N$ we must always have

$$(-1)^{Nk(E)} = (-1)^{-Nk(E)}. \tag{E.22}$$

Since $\overline{\mathcal{N}(E)}$ is a positive real number, any phase obtained from the integral $I_\Gamma$ must compensate for the one coming from $k(E)$. Therefore, with a bit of lack of rigor, we can conclude that for all physically meaningful values of $k(E)$, the following equalities hold:

$$e^{iNv(z_+)} = (-1)^{Nk(E)} = (-1)^{-Nk(E)} = e^{-iNv(z_+)}, \tag{E.23}$$

and we write

$$I_\Gamma \simeq (-1)^{Nk(E)} \exp\{N\Sigma(E)\}, \tag{E.24}$$

with

$$k(E) = \frac{p}{2\pi(p-1)} E\sqrt{E_{th}^2 - E^2} + \frac{1}{\pi} \arctan\left(\frac{\sqrt{E_{th}^2 - E^2}}{E}\right), \tag{E.25}$$

and $\Sigma(E) = \text{Re}[G(z_+(E))]$. Physically, having $k(E) > 0$ means that for $E > E_{th}$ the contribution to the integral in Eq. (E.6) is dominated by saddles having an extensive number of unstable direction: this is why in literature $E_{th}$ is referred to as the *energy threshold* where the local minima cease to dominate [10, 72].

For $0 < E < |E_{th}|$, the branch-cut crosses the integration path $\Gamma$ and we can not use the saddle-point method straightforwardly. Instead, we first notice that it exist two curve in the complex plane, $\gamma_+$ and $\gamma_-$, passing respectively through the saddle points $z_+(E)$ and $z_-(E)$ (both being maxima of $u(z)$ along such curves) and ending up in a point $z = x_1(E)$ on the branch-cut.

However, we observe that if we split $\Gamma$ in two curves, $\Gamma^+$ and $\Gamma^-$, defined in such a way that

$$
\begin{aligned}
I_{\Gamma^+} &= \int_{\Gamma^+} dz\, e^{NG(z)} \equiv \int_0^{i\infty} dz\, e^{NG(z)} - \int_0^{x_1(E)} dx\, e^{NG(x)}, \\
I_{\Gamma^-} &= \int_{\Gamma^-} dz\, e^{NG(z)} \equiv \int_{-i\infty}^0 dz\, e^{NG(z)} + \int_0^{x_1(E)} dx\, e^{NG(x)},
\end{aligned}
\tag{E.26}
$$

then we have that $I_\Gamma = I_{\Gamma^+} + I_{\Gamma^-}$ and that $\Gamma^+$ and $\Gamma^-$ can be respectively deformed on $\gamma^+$ and $\gamma^-$, as shown in Fig. 8-(c). In this way, we can use the saddle-point method to evaluate $I_{\Gamma^+}$ and $I_{\Gamma^-}$ separately. As the equalities $u(z_+) = u(z_-)$ and $v(z_+) = -v(z_-)$ hold like in the previous case, we find once again that $I_\Gamma \simeq (-1)^{Nk(E)} \exp\{N\Sigma(E)]\}$, with $k(E)$ given by Eq. (E.25), $\Sigma(E) = \text{Re}[G(z_+(E))]$ and for $E$ in the range $[0, |E_{th}|]$.

Finally, for $E > |E_{th}|$ both the $z_+(E)$ and $z_-(E)$ lie on the branch-cut, like in Fig. 8-(d). By some algebraic manipulations, one can show that $u(x, y)$ has an absolute maximum at the point $z_-(E)$ along the level curves $\gamma^+$ and $\gamma^-$ of $v(x, y)$ that intersect $z_-(E)$, while $u(x, y)$ has a minimum at the point $z_+(E)$ along the level curves of $v(x, y)$ that intersect $z_+(E)$. As done for the case $0 < E < |E_{th}|$ we divide once again $\Gamma$ in the curves $\Gamma^+$ and $\Gamma_-$, both ending in $z_-(E)$ and deform each of them respectively along $\gamma^+$ and $\gamma^-$ (see Fig. 8-(d)). By observing that $v(z) = \pm\pi$ respectively along $\gamma^+$ and $\gamma^-$, we conclude that $k(E) = 1$ in the energy range taken into exam, meaning that for $E > |E_{th}|$ the majority of stationary points of $V_J(\boldsymbol{\sigma})$ are local maxima. At the same time, the application of the Laplace method gives us:

$$
\Sigma(E) = \frac{z_+(E)^2}{p(p-1)} + \log(z_+(E) - pE) - E^2 - \frac{1}{2}\log\frac{p}{2} + \frac{1}{2}.
\tag{E.27}
$$

In summary, in the whole energy range studied we have

$$
\Sigma(E) = \frac{z(E)^2}{p(p-1)} + \log(z(E) - pE) - E^2 - \frac{1}{2}\log\frac{p}{2} + \frac{1}{2},
\tag{E.28}
$$

where $z(E) = z_+(E)$ if $E < |E_{th}|$ and $z(E) = z_-(E)$ if $E > |E_{th}|$, while the average stability index is given by

$$
k(E) = \begin{cases}
0, & \text{if } E < E_{th}, \\
\frac{p}{2\pi(p-1)} E \sqrt{E_{th}^2 - E^2} + \frac{1}{\pi}\arctan\left(\frac{\sqrt{E_{th}^2 - E^2}}{E}\right), & \text{if } |E| < |E_{th}|, \\
1, & \text{if } E > |E_{th}|,
\end{cases}
\tag{E.29}
$$

which is the result plotted in Figure 4 of the main text.

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
