# Peer review of "Probing chaos in the spherical $p$-spin glass model"

_SciPost Physics, doi:SciPost Phys. 15, 190 (2023)_

## Round 1 · Referee Report · Anonymous (Referee 1) · 2023-9-1

Report

I am satisfied with the authors' response to my comments, it is nice to see a rough agreement with Bera et al. However, I still think the level of impact is such that publication in SciPost Core would be more appropriate.

---

## Round 1 · Referee Report · Anonymous (Referee 4) · 2023-9-21

Strengths

1- very detailed numerical analysis 2- very detailed discussion of the results

Report

This paper presents a thorough numerical study of classical chaos in a p-spin glass spherical model, using the Truncated Wigner method as key tool. The authors investigate and discuss various indicators for chaos in this particular disordered system, such as the maximal Lyapunov exponent, the Kolmogorov-Sinai entropy, the fidelity susceptibility, as well as ergodicity as measured by the relaxation of temporal correlation functions. The main conclusion is that chaos is maximized at energies close to zero, which also corresponds to the part of the spectrum that features a maximum of complexity.

The paper is well written and I recommend it for publication in SciPost. I have two comments of minor importance that the authors may want to take into account:

1) The authors propose in Eq. (17) an expression that allows one to identify weak ergodicity breaking within the dynamical behaviour of the correlation function C (16), namely in terms of a long-time plateau level q_l towards which that correlation function relaxes in the limit of large time differences \tau. In Fig. 3(b) nonvanishing values for q_l are plotted for some negative values of the energy, based on the findings that are displayed in Fig. 3(a). Inspecting that latter viewgraph, however, it seems to me that also for all the negative values of the energy (except perhaps for the case E = -0.88) one has a relaxation of the correlation function towards zero, albeit on time scales \tau that exceed the window displayed in Fig. 3(a). To clarify this issue, I would propose to apply a more careful approach in order to determine q_l from the correlation function C (e.g. by plotting C as a function of 1 / \tau and extrapolating it to \tau^{-1} = 0).

2) In Eq. (12) the operators \hat{z}_i do not seem to be defined (at least I could not find their definition in the text). Do the authors mean \hat{\sigma}_i instead? Similarly, it seems to me that in Appendix D \hat{x}_i should read \hat{\sigma}_i.

  • validity: good
  • significance: good
  • originality: good
  • clarity: high
  • formatting: good
  • grammar: good

Author:  Lorenzo Correale  on 2023-10-18  [id 4043]

(in reply to Report 2 on 2023-09-21)
Category:
answer to question
correction

We thank the referee for her/his comments and suggestions, which we attempt to answer to in the following:

1) We agree with the referee regarding the need for a more meticulous analysis of the ergodicity breaking point. In our attempt to employ the method suggested by the referee, using the available data, a definitive transition did not unequivocally emerge. Consequently, the finite $q_1$ we computed should be viewed as an indication of 'slow' dynamics, where the thermalization time surpasses $t_{max}$, rather than a genuinely non-ergodic behavior. We have made this clarification in the revised version of the manuscript.

2) We apologize for the lack of clarity. The operator $\hat{\mathbf{z}}$ in Eq. (12) is a vector whose components are the $2N$ phase space operators $(\hat{\sigma}_1,\ldots,\hat{\sigma}_n,\hat{\Pi}_1,\ldots,\hat{\Pi}_N)$. We also noticed that we already used the letter ‘$z(t)$’ to identify the Lagrange multiplier, so we renamed the vector ‘$\hat{\mathbf{z}}$ ‘ as ‘$\hat{\mathbf{y}}$’ throughout the new version of the manuscript. We also fixed the typos in Appendix D noticed by the referee.

---

## Round 1 · Referee Report · Juan-Diego Urbina (Referee 2) · 2023-9-24

Report

I am happy with the way the authors addressed my comments/suggestions. I recommend publication.

---

## Round 1 · Author Response

Dear Editor,

We thank the referees for their report and for their observation that helped us to improve the terminology and make the approximations we used throughout the manuscript more transparent.
We have made revisions according to the referee suggestions, as reported in the "List of changes".
We hope that with these changes the paper will be accepted for publication in Scipost.

Sincerely,
The authors

---

## Round 1 · List of Changes

• Throughout the manuscript, we replaced the term "semi-classical" with "classical", whenever the second one is more appropriate for our discussion.

  • We improved the presentation of the model, explaining the historical root of the term "spin-glass", while also making a brief comparison between quantum fluctuations introduced by $\hbar$ in our model and the ones induced by a transverse field in more realistic spin $1/2$ systems. See the beginning of Section 2.

  • At the end of the first paragraph of Section 2, we discussed in more detail the results found in the previous work Phys. Rev. Lett. 128, 115302, in particular showing why the Lyapunov exponent computed therein does not contradict our findings and discussing in more detail the relation with the bound on chaos, proven in J. High Energ. Phys. 2016, 106. We also discussed why we can not compute the Lyapunov exponent at extremely low energies, at the end of Section 2.

  • At the end of Section 2, we also stated more clearly that the Wigner function we are using is not rigorous and discussed how our results could be improved by using a Wigner function representing realistic quantum state even at finite $\hbar$.

  • In Section 3, clarified in more detail the meaning of our computation for the asymptotic value $q_1$ of the correlation function, at finite $t_{max}$.

  • In Section 4 (page 12), we improved our discussion on the concomitance between maximal number of saddles and highest value of the Lyapunov exponent.

  • We included in Appendix C some details on the technical steps leading to eq.(21).

  • We fixed the typos noticed by the Referee 3.

  • We included the references suggested by the Referee 1.

---

## Round 2 · Author Response

Dear Editor,

We thank the referees for their report.
We have made revisions based on the observations of Referee 2, as reported in the "List of changes".
We hope that with these changes the paper will be accepted for publication in Scipost.

---

## Round 2 · List of Changes

• We fixed all the typos noticed by the referee and clarified the definition of the vector $\mathbf{z}$ (now renamed as "$\mathbf{y}$").

  • We improved the physical interpretation of the dynamics from the correlation functions, making a clear distinction between a real "ergodicity breaking" and "slow dynamics".

---

## Editorial Decision

published